# Achieving the Heisenberg limit in quantum metrology using quantum error correction

Sisi Zhou[1,2], Mengzhen Zhang[1,2], John Preskill[3] & Liang Jiang [1,2]

Quantum metrology has many important applications in science and technology, ranging from frequency spectroscopy to gravitational wave detection. Quantum mechanics imposes a fundamental limit on measurement precision, called the Heisenberg limit, which can be achieved for noiseless quantum systems, but is not achievable in general for systems subject to noise. Here we study how measurement precision can be enhanced through quantum error correction, a general method for protecting a quantum system from the damaging effects of noise. We find a necessary and sufficient condition for achieving the Heisenberg limit using quantum probes subject to Markovian noise, assuming that noiseless ancilla systems are available, and that fast, accurate quantum processing can be performed. When the sufficient condition is satisfied, a quantum error-correcting code can be constructed that suppresses the noise without obscuring the signal; the optimal code, achieving the best possible precision, can be found by solving a semidefinite program.

[1] Departments of Applied Physics and Physics, Yale University, New Haven, CT 06511, USA. [2] Yale Quantum Institute, Yale University, New Haven, CT 06520, USA. [3] Institute for Quantum Information and Matter, California Institute of Technology, Pasadena, CA 91125, USA. Correspondence and requests for materials should be addressed to S.Z. (email: sisi.zhou@yale.edu) or to L.J. (email: liang.jiang@yale.edu)

Quantum metrology concerns the task of estimating a parameter, or several parameters, characterizing the Hamiltonian of a quantum system. This task is performed by preparing a suitable initial state of the system, allowing it to evolve for a specified time, performing a suitable measurement, and inferring the value of the parameter(s) from the measurement outcome. Quantum metrology is of great importance in science and technology, with wide applications including frequency spectroscopy, magnetometry, accelerometry, gravimetry, gravitational wave detection, and other high-precision measurements[1–9].

Quantum mechanics places a fundamental limit on measurement precision, called the Heisenberg limit (HL), which constrains how the precision of parameter estimation improves as the total probing time $t$ increases. According to HL, the scaling of precision with $t$ can be no better than $1/t$; equivalently, precision scales no better than $1/N$ with the total number of probes $N$ used in an experiment. For a noiseless system, HL scaling is attainable in principle by, for example, preparing an entangled "cat" state of $N$ probes[10–12]. In practice, though, in most cases environmental decoherence imposes a more severe limitation on precision; instead of HL, precision scales like $1/\sqrt{N}$, called the standard quantum limit (SQL), which can be achieved by using $N$ independent probes[13–18]. The quest for measurement schemes surpassing the SQL has inspired a variety of clever strategies, such as squeezing the vacuum[1], optimizing the probing time[19], monitoring the environment[20,21], and exploiting non-Markovian effects[22–24].

Quantum error correction (QEC) is a particularly powerful tool for enhancing the precision of quantum metrology[25–30]. Quantum error correction is a method for reducing noise in quantum channels and quantum processors[31–33]. In principle, it enables a noisy quantum computer to simulate faithfully an ideal quantum computer, with reasonable overhead cost, if the noise is not too strong or too strongly correlated. But the potential value of QEC in quantum metrology has not yet been fully fleshed out, even as a matter of principle. A serious obstacle for applications of QEC to sensing is that it may in some cases be exceedingly hard to distinguish the signal arising from the Hamiltonian evolution of the probe system from the effects of the noise acting on the probe. Nevertheless, it has been shown that QEC can be invoked to achieve HL scaling under suitable conditions[25–28], and experiments demonstrating the efficacy of QEC in a room-temperature hybrid spin register have recently been conducted[34].

As is the case for quantum computing, we should expect positive (or negative) statements about improving metrology via QEC to be premised on suitable assumptions about the properties of the noise and the capabilities of our quantum hardware. But what assumptions are appropriate, and what can be inferred from these assumptions? In this paper, we assume that the probes used for parameter estimation are subject to noise described by a Markovian master equation[35,36], where the strength and structure of this noise is beyond the experimentalist's control. However, aside from the probe system, the experimentalist also has noiseless ancilla qubits at her disposal, and the ability to apply noiseless quantum gates that act jointly on the ancilla and probe; she can also perform perfect ancilla measurements, and reset the ancillas after measurement. Furthermore, we assume that a quantum gate or measurement can be executed in an arbitrarily short time (though the Markovian description of the probe's noise is assumed to be applicable no matter how fast the processing).

Previous studies have shown that whether HL scaling can be achieved by using QEC to protect a noisy probe depends on the algebraic structure of the noise. For example, if the probe is a qubit (two-dimensional quantum system), then HL scaling is possible when detecting a $\sigma_z$ signal in the presence of bit-flip ($\sigma_x$) errors[25–28], but not for dephasing ($\sigma_z$) noise acting on the probe, even if arbitrary quantum controls and feedback are allowed[16]. (Here $\sigma_{x,y,z}$ denote the Pauli matrices.) For this example, we say that $\sigma_x$ noise is "perpendicular" to the $\sigma_z$ signal, while $\sigma_z$ noise is "parallel" to the signal. In some previous work on improving metrology using QEC, perpendicular noise has been assumed[25,26], but this assumption is not necessary—for a qubit probe, HL scaling is achievable for any noise channel with just one Hermitian jump operator $L$, except in the case where the signal Hamiltonian $H$ commutes with $L$[37].

In this paper, we extend these results to any finite-dimensional probe, finding the necessary and sufficient condition on the noise for achievability of HL scaling. This condition is formulated as an algebraic relation between the signal Hamiltonian whose coefficient is to be estimated and the Lindblad operators $\{L_k\}$ that appear in the master equation describing the evolution of the probe. We prove that (1) if the signal Hamiltonian can be expressed as a linear combination of the identity operator $I$, the Lindblad operators $L_k$, their Hermitian conjugates $L_k^\dagger$ and the products $L_k^\dagger L_j$ for all $k$, $j$, then SQL scaling cannot be surpassed. (2) Otherwise HL scaling is achievable by using a QEC code such that the effective "logical" evolution of the probe is noiseless and unitary. Notably, under the assumptions considered here, either SQL scaling cannot be surpassed or HL scaling is achievable via quantum coding; in contrast, intermediate scaling is possible in some other metrology scenarios[19]. For the case where our sufficient condition is satisfied, we explicitly construct a QEC code that achieves HL scaling. Furthermore, we show that searching for the QEC code that achieves optimal precision can be formulated as a semidefinite program (SDP) that can be efficiently solved numerically, and can be solved analytically in some special cases. Our sufficient condition cannot be satisfied if the noise channel is full rank, and is therefore not applicable for generic noise. However, for noise which is $\epsilon$-close to meeting our criterion, using the QEC code ensures that HL scaling can be maintained approximately for a time $O(1/\epsilon)$, before crossing over to asymptotic SQL scaling.

## Results

**Sequential scheme for quantum metrology.** We assume that the probes used for parameter estimation are subject to noise described by a Markovian master equation. In addition to the probe system, the experimentalist also has noiseless ancilla qubits at her disposal. She can apply fast, noiseless quantum gates that act jointly on the ancilla and probe; she can also perform perfect ancilla measurements, and reset the ancillas after measurement.

We endow the experimentalist with these powerful tools because we wish to address, as a matter of principle, how effectively QEC can overcome the deficiencies of the noisy probe system. Our scenario may be of practical interest as well, in hybrid quantum systems where ancillas are available, which have a much longer coherence time than the probe. For example, sensing of a magnetic field with a probe electron spin can be enhanced by using a quantum code, which takes advantage of the long coherence time of a nearby (ancilla) nuclear spin in diamond[34]. In cases where noise acting on the ancilla is weak but not completely negligible, we may be able to use QEC to enhance the coherence time of the ancilla, thus providing better justification for our idealized setting in which the ancilla is effectively noiseless. Our assumption that quantum processing is much faster than characteristic decoherence rates is necessary for QEC to succeed in quantum computing as well as in quantum metrology, and recent experimental progress indicates that this assumption is applicable in at least some realistic settings. For example, in superconducting devices, QEC has reached the break-

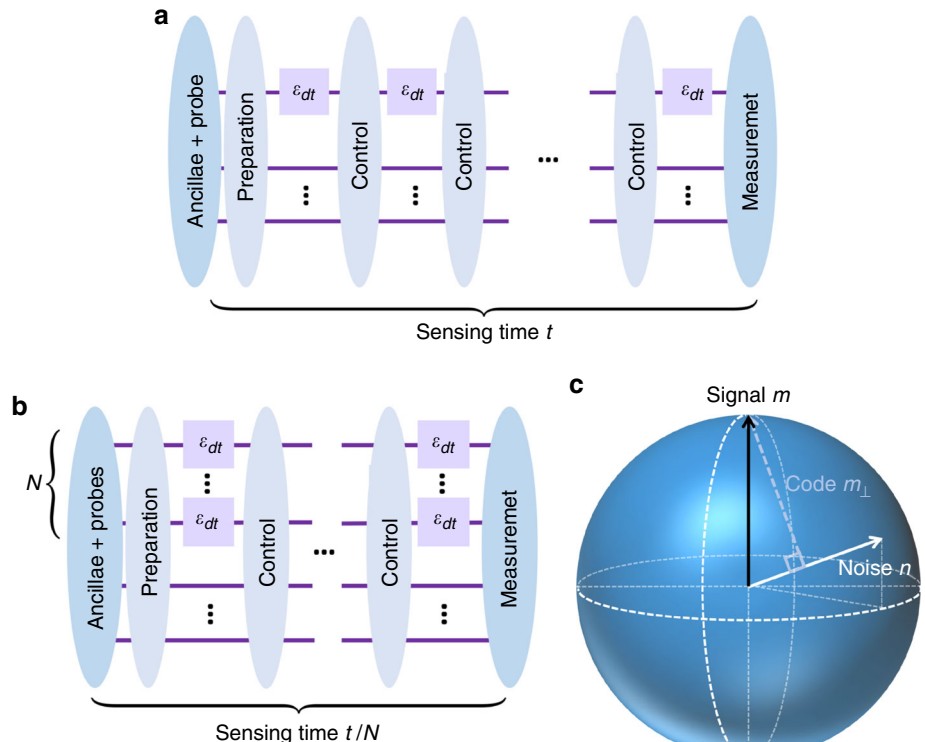

**Fig. 1** Metrology schemes and qubit probe. **a** The sequential scheme. One probe sequentially senses the parameter for time $t$, with quantum controls applied every $dt$. **b** The parallel scheme. $N$ probes sense the parameter for time $t/N$ in parallel. The parallel scheme can be simulated by the sequential scheme. **c** The relation between the signal Hamiltonian, the noise, and the QEC code on the Bloch sphere for a qubit probe

even point where the lifetime of an encoded qubit exceeds the natural lifetime of the constituents of the system;[38] one- and two-qubit logical operations have also been demonstrated[39,40]. Moreover, if sensing could be performed using a probe encoded within a noiseless subspace or subsystem[41], then active error correction would not be needed to protect the probe, making the QEC scheme more feasible using near-term technology.

In accord with our assumptions, we adopt the sequential scheme for quantum metrology[37,42,43] (Fig. 1a). In this scheme, a single noisy probe senses the unknown parameter for many rounds, where each round lasts for a short time interval $dt$, and the total number of rounds is $t/dt$, where $t$ is the total sensing time. In between rounds, an arbitrary (noiseless) quantum operation can be applied instantaneously, which acts jointly on the probe and the noiseless ancillas. The rapid operations between rounds empower us to perform QEC, suppressing the damaging effects of the noise on the probe. Note that this sequential scheme can simulate a parallel scheme (Fig. 1b), in which $N$ probes simultaneously sense the parameter for time $t/N$[37,42].

**Necessary and sufficient condition for HL.** We denote the $d$-dimensional Hilbert space of our probe by $\mathcal{H}_P$, and we assume the state $\rho_P$ of the probe evolves according to a time-homogeneous Lindblad master equation of the form (with $\hbar = 1$)[31,35,36],

$$\frac{d\rho_P}{dt} = -i\left[H, \rho_P\right] + \sum_{k=1}^{r}\left(L_k\rho_P L_k^\dagger - \tfrac{1}{2}\left\{L_k^\dagger L_k, \rho_P\right\}\right), \quad (1)$$

where $H$ is the probe's Hamiltonian, $\{L_k\}$ are the Lindblad jump operators, and $r$ is the "rank" of the noise channel acting on the probe (the smallest number of Lindblad operators needed to describe the channel). The Hamiltonian $H$ depends on a parameter $\omega$, and our goal is to estimate $\omega$. For simplicity, we will assume that $H = \omega G$ is a linear function of $\omega$, but our arguments

actually apply more generally. If $H(\omega)$ is not a linear function of $\omega$, the coding scheme we describe below can be repeated many times if necessary, using our latest estimate of $\omega$ after each round to adjust the scheme used in the next round. By including in the protocol an inverse Hamiltonian evolution step $\exp(iH(\hat{\omega})dt)$ applied to the probe, where $\hat{\omega}$ is the estimated value of $\omega$, we can justify the linear approximation when $\hat{\omega}$ is sufficiently accurate. The asymptotic scaling of precision with the total probing time is not affected by the preliminary adaptive rounds[44].

We denote by $\mathcal{H}_A$ the $d$-dimensional Hilbert space of a noiseless ancilla system, whose evolution is determined solely by our fast and accurate quantum controls. Over the small time interval $dt$, during which no controls are applied, the ancilla evolves trivially, and the joint state $\rho$ of probe and ancilla evolves according to the quantum channel:

$$\mathcal{E}_{dt}(\rho) = \rho - i\omega[G, \rho]dt$$
$$+ \sum_{k=1}^{r}\left(L_k\rho L_k^\dagger - \tfrac{1}{2}\left\{L_k^\dagger L_k, \rho\right\}\right)dt + O(dt^2), \quad (2)$$

where $G$, $L_k$ are shorthand for $G \otimes I$, $L_k \otimes I$, respectively. We assume that this time interval $dt$ is sufficiently small that corrections higher order in $dt$ can be neglected. In between rounds of sensing, each lasting for time $dt$, control operations acting on $\rho$ are applied instantaneously.

Our conclusions about HL and SQL scaling of parameter estimation make use of an algebraic condition on the master equation that we will refer to often, and it will therefore be convenient to have a name for this condition. We will call it the Hamiltonian-not-in-Lindblad span (HNLS) condition, or simply HNLS, an acronym for "Hamiltonian-not-in-Lindblad span." We denote by $\mathcal{S}$ the linear span of the operators $I$, $L_k$, $L_k^\dagger$, $L_k^\dagger L_j$ (for all $k$ and $j$ ranging from 1 to $r$), and say that the Hamiltonian $H$ obeys the HNLS condition if $H$ is not contained in $\mathcal{S}$. Now we can

state our main conclusion about parameter estimation using fast and accurate quantum controls as Theorem 1.

*Theorem 1:* Consider a finite-dimensional probe with Hamiltonian $H = \omega G$, subject to Markovian noise described by a Lindblad master equation with jump operators $\{L_k\}$. Then $\omega$ can be estimated with HL (Heisenberg-limited) precision if and only if $G$ and $\{L_k\}$ obey the HNLS (Hamiltonian-not-in-Lindblad-span) condition.

Theorem 1 applies if the ancilla is noiseless, and also for an ancilla subject to Markovian noise obeying suitable conditions, as we discuss in the Methods.

**Qubit probe.** To illustrate how Theorem 1 works, let's look at the case where the probe is a qubit, which has been discussed in detail in ref. [37]. Suppose one of the Lindblad operators is $L_1 \propto \mathbf{n} \cdot \sigma$, where $\mathbf{n} = \mathbf{n_r} + i\mathbf{n_i}$ is a normalized complex 3-vector and $\mathbf{n_r}$, $\mathbf{n_i}$ are its real and imaginary parts, so that $L_1^\dagger L_1 \propto (\mathbf{n}^* \cdot \sigma)(\mathbf{n} \cdot \sigma) = I + 2(\mathbf{n_i} \times \mathbf{n_r}) \cdot \sigma$. If $\mathbf{n_r}$ and $\mathbf{n_i}$ are not parallel vectors, then $\mathbf{n_r}$, $\mathbf{n_i}$, and $\mathbf{n_i} \times \mathbf{n_r}$ are linearly independent, which means that $I$, $L_1$, $L_1^\dagger$, and $L_1^\dagger L_1$ span the four-dimensional space of linear operators acting on the qubit. Hence HNLS cannot be satisfied by any qubit Hamiltonian, and therefore parameter estimation with HL scaling is not possible according to Theorem 1. We conclude that for HL scaling to be achievable, $\mathbf{n_r}$ and $\mathbf{n_i}$ must be parallel, which means that (after multiplying $L_1$ by a phase factor if necessary) we can choose $L_1$ to be Hermitian[37]. Moreover, if $L_1$ and $L_2$ are two linearly independent Hermitian traceless Lindblad operators, then $\{I, L_1, L_2, L_1 L_2\}$ span the space of qubit linear operators and HL scaling cannot be achieved. In fact, for a qubit probe, HNLS can be satisfied only if there is a single Hermitian (not necessarily traceless) Lindblad operator $L$, and the Hamiltonian does not commute with $L$.

We will describe below how to achieve HL scaling for any master equation that satisfies HNLS, by constructing a two-dimensional QEC code that protects the probe from the Markovian noise. To see how the code works for a qubit probe, suppose $G = \frac{1}{2}\mathbf{m} \cdot \sigma$ and $L \propto \mathbf{n} \cdot \sigma$, where $\mathbf{m}$ and $\mathbf{n}$ are unit vectors in $\mathbb{R}^3$ (Fig. 1c). Then the basis vectors for the QEC code may be chosen to be:

$$|C_0\rangle = |\mathbf{m}_\perp, +\rangle_P \otimes |0\rangle_A, \quad |C_1\rangle = |\mathbf{m}_\perp, -\rangle_P \otimes |1\rangle_A; \quad (3)$$

here $|0\rangle_A$, $|1\rangle_A$ are basis states for the ancilla qubit, and $|\mathbf{m}_\perp, \pm\rangle_P$ are the eigenstates with eigenvalues $\pm 1$ of $\mathbf{m}_\perp \cdot \sigma$ where $\mathbf{m}_\perp$ is the (normalized) component of $\mathbf{m}$ perpendicular to $\mathbf{n}$. In particular, if $\mathbf{m} \perp \mathbf{n}$ (perpendicular noise), then $|C_0\rangle = |\mathbf{m}, +\rangle_P \otimes |0\rangle_A$ and $|C_1\rangle = |\mathbf{m}, -\rangle_P \otimes |1\rangle_A$, the coding scheme previously discussed in refs. [25–28].

In the case of perpendicular noise, we estimate $\omega$ by tracking the evolution in the code space of a state initially prepared as (in a streamlined notation) $\psi(0) = (|+, 0\rangle + |-, 1\rangle)/\sqrt{2}$; neglecting the noise, this state evolves in time $t$ to

$$|\psi(t)\rangle = \frac{1}{\sqrt{2}}\left(e^{-i\omega t/2}|+, 0\rangle + e^{i\omega t/2}|-, 1\rangle\right). \quad (4)$$

If a jump then occurs at time $t$, the state is transformed to

$$|\psi'(t)\rangle = \frac{1}{\sqrt{2}}\left(e^{-i\omega t/2}|-, 0\rangle + e^{i\omega t/2}|+, 1\rangle\right). \quad (5)$$

Jumps are detected by performing a two-outcome measurement that projects onto either the span of $\{|+, 0\rangle, |-, 1\rangle\}$ (the code space) or the span of $\{|-, 0\rangle, |+, 1\rangle\}$ (orthogonal to the code space), and when detected they are immediately corrected by flipping the probe. Because errors are immediately corrected, the error-corrected evolution matches perfectly the ideal evolution (without noise), for which HL scaling is possible.

When the noise is not perpendicular to the signal, then not just the jumps but also the Hamiltonian evolution can rotate the joint state of probe and ancilla away from the code space. However, after evolution for the short time interval $dt$, the overlap with the code space remains large, so that the projection onto the code space succeeds with probability $1 - O(dt^2)$. Neglecting $O(dt^2)$ corrections, then, the joint probe-ancilla state rotates noiselessly in the code space, at a rate determined by the component of the Hamiltonian evolution along the code space. As long as this component is nonzero, HL scaling can be achieved.

We will see that this reasoning can be extended to any finite-dimensional probe satisfying HNLS, including quantum many-body systems and (appropriately truncated) bosonic channels. Here we briefly mention a few other cases where HNLS applies, and therefore HL scaling is achievable. (1) For a many-qubit system, suppose that each Lindblad jump operator $L_k$ is supported on no more than $t$ qubits (hence each $L_k^\dagger L_j$ is supported on no more than $2t$ qubits), and the Hamiltonian contains at least one term acting on at least $2t + 1$ qubits. Then HNLS holds. (2) Consider a $d$-dimensional system (a qudit), and define generalized Pauli operators

$$X = \sum_{k=0}^{d-1} |k+1\rangle\langle k|, \quad Z = \sum_{k=0}^{d-1} e^{2\pi i k/d}|k\rangle\langle k|, \quad (6)$$

(where addition is modulo $d$). Suppose that the Hamiltonian $H$ ($Z$) is a non-constant function of $Z$ and that there is a single Lindblad jump operator $L(X)$ which is a function of $X$. Then HNLS holds. HNLS may also apply for a multi-qubit sensor with qubits at distinct spatial positions, where the signal and noise are parallel for each individual qubit, but the signal and noise depend on position in different ways[45].

We must explain how, when HNLS holds, a quantum code can be constructed that achieves HL scaling. But first we will discuss why HL is impossible when HNLS fails.

**Non-achievability of HL when HNLS fails.** The necessary condition for HL scaling can be derived from the quantum Cramér–Rao bound[46–48]

$$\delta\hat{\omega} \geq 1/\sqrt{R \cdot \mathcal{F}(\rho_\omega(t))}; \quad (7)$$

here $\hat{\omega}$ denotes any unbiased estimator for the parameter $\omega$, and $\delta\hat{\omega}$ is that estimator's standard deviation. $\mathcal{F}(\rho_\omega(t))$ is the quantum Fisher information (QFI) of the state $\rho_\omega(t)$; this state is obtained by preparing an initial state $\rho_{in}$ of the probe, and then evolving this state for total time $t$, where the evolution is governed by the $\omega$-dependent probe Hamiltonian $H(\omega)$, the Markovian noise acting on the probe, and our fast quantum controls. For a scheme in which the measurement protocol is repeated many times in succession, $R$ denotes the number of such repetitions. Here we show that $\mathcal{F}(\rho_\omega(t))$ is at most asymptotically linear in $t$ when the Hamiltonian $H(\omega)$ is contained in the linear span (denoted $\mathcal{S}$) of $I$, $L_k$, $L_k^\dagger$, and $L_k^\dagger L_j$, which means that SQL scaling cannot be surpassed in this case.

Though it is challenging to compute the maximum attainable QFI for arbitrary quantum channels, useful upper bounds on QFI can be derived, which provide lower bounds on the precision of quantum metrology[15–18,37,42,49]. The quantum channel describing the joint evolution of probe and ancilla has a Kraus operator representation

$$\mathcal{E}_{dt}(\rho) = \sum_k K_k \rho K_k^\dagger, \quad (8)$$

and in terms of these Kraus operators we define

$$\alpha_{dt} = \sum_k \dot{K}_k^\dagger \dot{K}_k = \dot{\mathbf{K}}^\dagger \dot{\mathbf{K}}, \qquad (9)$$

$$\beta_{dt} = i \sum_k \dot{K}_k^\dagger K_k = i \dot{\mathbf{K}}^\dagger \mathbf{K}, \qquad (10)$$

where we express the Kraus operators in vector notation $\mathbf{K} := (K_0, K_1, \dots)^T$, and the over-dot means the derivative with respect to $\omega$. If $\rho_{\mathrm{in}}$ is the initial joint state of probe and ancilla at time 0, and $\rho(t)$ is the corresponding state at time $t$, then the upper bound on the QFI

$$\mathcal{F}(\rho(t)) \le 4\frac{t}{dt}\|\alpha_{dt}\| + 4\left(\frac{t}{dt}\right)^2 \|\beta_{dt}\|\left(\left(\|\beta_{dt}\| + 2\sqrt{\|\alpha_{dt}\|}\right)\right) \quad (11)$$

($\|\cdot\|$ denotes the operator norm) derived by the "channel extension method" holds for any choice of $\rho_{\mathrm{in}}$ even when fast and accurate quantum controls are applied during the evolution[37]. This upper bound on the QFI provides a lower bound on the precision $\delta\hat{\omega}$ via Eq. (7).

Kraus representations are not unique—for any matrix $u$ satisfying $u^\dagger u = I$, $\mathbf{K}' = u\mathbf{K}$ represents the same channel as $\mathbf{K}$. Hence, we can tighten the upper bound on the QFI by minimizing the RHS of Eq. (11) over all such valid Kraus representations. We see that

$$\dot{\mathbf{K}}' = u(\dot{\mathbf{K}} - ih\mathbf{K}), \quad \dot{\mathbf{K}}'^\dagger = (\dot{\mathbf{K}} - ih\mathbf{K})^\dagger u^\dagger \quad (12)$$

where $h = iu^\dagger \dot{u}$. Therefore, to find $\alpha_{dt}$ and $\beta_{dt}$ providing the tightest upper bound on the QFI, it suffices to replace $\dot{\mathbf{K}}$ by $\dot{\mathbf{K}} - ih\mathbf{K}$ and to optimize over the Hermitian matrix $h$.

To evaluate the bound for asymptotically large $t$, we expand $\alpha_{dt}, \beta_{dt}, h$ in powers of $\sqrt{dt}$:

$$\alpha_{dt} = \alpha^{(0)} + \alpha^{(1)}\sqrt{dt} + \alpha^{(2)}dt + O\left(dt^{3/2}\right), \quad (13)$$

$$\beta_{dt} = \beta^{(0)} + \beta^{(1)}\sqrt{dt} + \beta^{(2)}dt + \beta^{(3)}dt^{3/2} + O\left(dt^2\right), \quad (14)$$

$$h = h^{(0)} + h^{(1)}\sqrt{dt} + h^{(2)}dt + h^{(3)}dt^{3/2} + O\left(dt^2\right). \quad (15)$$

We show in the Methods that the first two terms in $\alpha_{dt}$ and the first four terms in $\beta_{dt}$ can all be set to 0 by choosing a suitable $h$, assuming that HNLS is violated. We therefore have $\alpha_{dt} = O(dt)$ and $\beta_{dt} = O(dt^2)$, so that the second term in the RHS of Eq. (11) vanishes as $dt \to 0$:

$$\mathcal{F}(\rho(t)) \le 4\|\alpha^{(2)}\|t, \quad (16)$$

proving that SQL scaling cannot be surpassed when HNLS is violated (the necessary condition in Theorem 1). We require the probe to be finite dimensional in the statement of Theorem 1 because otherwise the norm of $\alpha_{dt}$ or $\beta_{dt}$ could be infinite. The theorem can be applied to the case of a probe with an infinite-dimensional Hilbert space if the state of the probe is confined to a finite-dimensional subspace even for asymptotically large $t$.

**QEC code for HL scaling when HNLS holds**. To prove the sufficient condition for HL scaling, we show that a QEC code achieving HL scaling can be explicitly constructed if $H(\omega)$ is not in the linear span $\mathcal{S}$. Our discussion of the qubit probe indicates how a QEC code can be used to achieve HL scaling for estimating the parameter $\omega$. The code allows us to correct quantum jumps whenever they occur, and in addition the noiseless error-corrected evolution in the code space depends nontrivially on $\omega$. Similar considerations apply to higher-dimensional probes. Let

$\Pi_C$ denote the projection onto the code space. Jumps are correctable if the code satisfies the error correction conditions[31–33], namely:

$$[1]\ \Pi_C L_k \Pi_C = \lambda_k \Pi_C, \ \forall k, \quad (17)$$

$$[2]\ \Pi_C L_k^\dagger L_j \Pi_C = \mu_{kj} \Pi_C, \ \forall k, j, \quad (18)$$

for some complex numbers $\lambda_k$ and $\mu_{kj}$. The error-corrected joint state of probe and ancilla evolves according to the unitary channel (asymptotically)

$$\frac{d\rho}{dt} = -i[H_{\mathrm{eff}}, \rho] \quad (19)$$

where $H_{\mathrm{eff}} = \Pi_C H \Pi_C = \omega G_{\mathrm{eff}}$. There is a code state for which the evolution depends nontrivially on $\omega$ provided that

$$[3]\ \Pi_C G \Pi_C \ne \text{constant} \ \Pi_C. \quad (20)$$

For this noiseless evolution with effective Hamiltonian $\omega G_{\mathrm{eff}}$, the QFI of the encoded state at time $t$ is

$$\mathcal{F}(\rho(t)) = 4t^2\left[\mathrm{tr}\left(\rho_{\mathrm{in}} G_{\mathrm{eff}}^2\right) - \left(\mathrm{tr}\left(\rho_{\mathrm{in}} G_{\mathrm{eff}}\right)\right)^2\right], \quad (21)$$

where $\rho_{\mathrm{in}}$ is the initial state at time $t = 0$. The QFI is maximized by choosing the initial pure state

$$|\psi_{\mathrm{in}}\rangle = \frac{1}{\sqrt{2}}\left(|\lambda_{\min}\rangle + |\lambda_{\max}\rangle\right), \quad (22)$$

where $|\lambda_{\min}\rangle, |\lambda_{\max}\rangle$ are the eigenstates of $G_{\mathrm{eff}}$ with the minimal and maximal eigenvalues; with this choice the QFI is

$$\mathcal{F}(\rho(t)) = t^2(\lambda_{\max} - \lambda_{\min})^2. \quad (23)$$

By measuring in the appropriate basis at time $t$, we can estimate $\omega$ with a precision that saturates the Cramér–Rao bound in the asymptotic limit of a large number of measurements, hence realizing HL scaling.

To prove the sufficient condition in Theorem 1, we will now show that a code with properties (1)–(3) can be constructed whenever HNLS is satisfied. (For further justification of these conditions see the Methods.) In this code construction we make use of a noiseless ancilla system, but as we discuss in the Methods, the construction can be extended to the case where the ancilla system is subject to Markovian noise obeying suitable conditions.

To see how the code is constructed, note that the $d$-dimensional Hermitian matrices form a real Hilbert space where the inner product of two matrices $A$ and $B$ is defined to be $\mathrm{tr}(AB)$. Let $\mathcal{S}$ denote the subspace of Hermitian matrices spanned by $I$, $L_k + L_k^\dagger$, $i\left(L_k - L_k^\dagger\right)$, $L_k^\dagger L_j + L_j^\dagger L_k$, and $i\left(L_k^\dagger L_j - L_j^\dagger L_k\right)$ for all $k, j$. Then $G$ has a unique decomposition into $G = G_\parallel + G_\perp$, where $G_\parallel \in \mathcal{S}$ and $G_\perp \perp \mathcal{S}$.

If HNLS holds, then $G_\perp$ is nonzero. It must also be traceless, in order to be orthogonal to $I$, which is contained in $\mathcal{S}$. Therefore, using the spectral decomposition, we can write $G_\perp = \frac{1}{2}\left(\mathrm{tr}|G_\perp|\right)(\rho_0 - \rho_1)$, where $\rho_0$ and $\rho_1$ are trace-one positive matrices with orthogonal support and $|G_\perp| := \sqrt{G_\perp^2}$. Our QEC code is chosen to be the two-dimensional subspace of $\mathcal{H}_P \otimes \mathcal{H}_A$ spanned by $|C_0\rangle$ and $|C_1\rangle$, which are normalized purifications of $\rho_0$ and $\rho_1$ respectively, with orthogonal support in $\mathcal{H}_A$. (If the probe is $d$-dimensional, a $d$-dimensional ancilla can purify its state.) Because the code basis states have orthogonal support on $\mathcal{H}_A$, it follows that, for any $O$ acting on $\mathcal{H}_P$,

$$\langle C_0| O \otimes I |C_1\rangle = 0 = \langle C_1| O \otimes I |C_0\rangle, \quad (24)$$

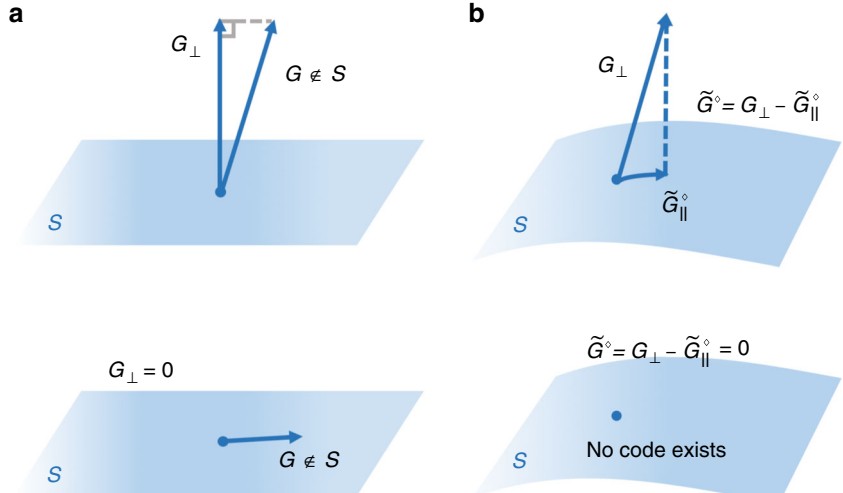

**Fig. 2** Schematic illustration of HNLS and code optimization. **a** $G_\perp$ is the projection of $G$ onto $\mathcal{S}$ in the Hilbert space of Hermitian matrices equipped with the Hilbert-Schmidt norm $\sqrt{\mathrm{tr}(O \cdot O)}$. $G_\perp \neq 0$ if and only if $G \notin \mathcal{S}$, which is the HNLS condition. **b** $\tilde{G}^\diamond$ is the projection of $G$ onto $\mathcal{S}$ in the linear space of Hermitian matrices equipped with the operator norm $\|O\| = \max_{|\psi\rangle}\langle\psi|O|\psi\rangle$. In general, the optimal QEC code can be contructed from $\tilde{G}^\diamond$ and $\tilde{G}^\diamond$ is not necessarily equal to $G_\perp$

and furthermore

$$\mathrm{tr}((|C_0\rangle\langle C_0| - |C_1\rangle\langle C_1|)(O \otimes I))$$
$$= \mathrm{tr}\left((\rho_0 - \rho_1)O\right) = \frac{2\,\mathrm{tr}(G_\perp O)}{\mathrm{tr}|G_\perp|}. \quad (25)$$

In particular, for any $O$ in the span $\mathcal{S}$ we have $\mathrm{tr}(G_\perp O) = 0$, and therefore

$$\langle C_0|(O \otimes I)|C_0\rangle = \langle C_1|(O \otimes I)|C_1\rangle. \quad (26)$$

Code properties (1)–(3) now follow from Eqs. (24) and (26). For this two-dimensional code, the projector onto the code space is

$$\Pi_C = |C_0\rangle\langle C_0| + |C_1\rangle\langle C_1|, \quad (27)$$

and therefore

$$\Pi_C(O \otimes I)\Pi_C = \langle C_0|(O \otimes I)|C_0\rangle\Pi_C \quad (28)$$

for $O \in \mathcal{S}$, which implies properties (1) and (2) because $L_k$ and $L_k^\dagger L_j$ are in $\mathcal{S}$. Property (3) is also satisfied by the code, because $\langle C_0|G|C_0\rangle - \langle C_1|G|C_1\rangle = 2\,\mathrm{tr}(G_\perp^2)/\mathrm{tr}|G_\perp| > 0$, which means that the diagonal elements of $\Pi_C G \Pi_C$ are not equal when projected onto the code space. Thus, we have demonstrated the existence of a code with properties (1) and (3).

**Code optimization.** When HNLS is satisfied, we can use our QEC code, along with fast and accurate quantum control, to achieve noiseless evolution of the error-corrected probe, governed by the effective Hamiltonian $H_{\mathrm{eff}} = \Pi_C H \Pi_C = \omega G_{\mathrm{eff}}$ where $\Pi_C$ is the orthogonal projection onto the code space. Because the optimal initial state Eq. (22) is a superposition of just two eigenstates of $G_{\mathrm{eff}}$, a two-dimensional QEC code suffices for achieving the best possible precision. For a code with basis states $\{|C_0\rangle, |C_1\rangle\}$, the effective Hamiltonian is

$$G_{\mathrm{eff}} = |C_0\rangle\langle C_0|G_\perp|C_0\rangle\langle C_0| + |C_1\rangle\langle C_1|G_\perp|C_1\rangle\langle C_1|; \quad (29)$$

here we have ignored the contribution due to $G_\parallel$, which is an irrelevant additive constant if the code satisfies condition (2). We

have seen how to construct a code for which

$$\lambda_{\max} - \lambda_{\min} = 2\frac{\mathrm{tr}(G_\perp^2)}{\mathrm{tr}|G_\perp|}. \quad (30)$$

It is possible, though, that a larger value of this difference of eigenvalues could be achieved using a different code, improving the precision by a constant factor (independent of the time $t$).

To search for a better code, with basis states $\{|C_0\rangle, |C_1\rangle\}$, define

$$\tilde{\rho}_0 = \mathrm{tr}_A(|C_0\rangle\langle C_0|), \quad \tilde{\rho}_1 = \mathrm{tr}_A(|C_1\rangle\langle C_1|), \quad (31)$$

and consider

$$\tilde{G} = \tilde{\rho}_0 - \tilde{\rho}_1. \quad (32)$$

Conditions (1)–(2) on the code imply

$$\mathrm{tr}(\tilde{G}O) = 0, \ \forall O \in \mathcal{S}, \quad (33)$$

and we want to maximize

$$\lambda_{\max} - \lambda_{\min} = \mathrm{tr}(G_{\mathrm{eff}}\tilde{G}) = \mathrm{tr}(G_\perp\tilde{G}), \quad (34)$$

over matrices $\tilde{G}$ of the form Eq. (32) subject to Eq. (33). Note that $\tilde{G}$ is the difference of two normalized density operators, and therefore satisfies $\mathrm{tr}|\tilde{G}| \leq 2$. In fact, though, if $\tilde{G}$ obeys the constraint Eq. (33), then the constraint is still satisfied if we rescale $\tilde{G}$ by a real constant greater than one, which increases $\mathrm{tr}(G_\perp\tilde{G})$; hence the maximum of $\mathrm{tr}(G_\perp\tilde{G})$ is achieved for $\mathrm{tr}|\tilde{G}| = 2$, which means that $\tilde{\rho}_0$ and $\tilde{\rho}_1$ have orthogonal support.

Now recall that $G_\perp = \frac{1}{2}(\mathrm{tr}|G_\perp|)(\rho_0 - \rho_1)$ is also (up to normalization) a difference of density operators with orthogonal support, and obeys the constraint Eq. (33). The quantity to be maximized is proportional to

$$\mathrm{tr}[(\rho_0 - \rho_1)(\tilde{\rho}_0 - \tilde{\rho}_1)] = \mathrm{tr}(\rho_0\tilde{\rho}_0 + \rho_1\tilde{\rho}_1 - \rho_0\tilde{\rho}_1 - \rho_1\tilde{\rho}_0). \quad (35)$$

If $\rho_0$ and $\rho_1$ are both rank 1, then the maximum is achieved by choosing $\tilde{\rho}_0 = \rho_0$ and $\tilde{\rho}_1 = \rho_1$. Conditions (1)–(2) are satisfied by choosing $|C_0\rangle$ and $|C_1\rangle$ to be purifications of $\rho_0$ and $\rho_1$ with orthogonal support on $\mathcal{H}_A$. Thus, we have recovered the code we constructed previously. If $\rho_0$ or $\rho_1$ is higher rank, though, then a different code achieves a higher maximum, and hence better precision for parameter estimation.

**Geometrical picture.** There is an alternative description of the code optimization, with a pleasing geometrical interpretation. As discussed in the Methods, the optimization can be formulated as a SDP with a feasible dual program. By solving the dual program we find that, for the optimal QEC code, the QFI is

$$\mathcal{F}(\rho(t)) = 4t^2 \min_{\tilde{G}_\parallel \in \mathcal{S}} \left\| G_\perp - \tilde{G}_\parallel \right\|^2, \tag{36}$$

where $\|\cdot\|$ denotes the operator norm. In this sense, the QFI is determined by the minimal distance between $G_\perp$ and $\mathcal{S}$ (Fig. 2b).

We can recover the solution to the primal problem from the solution to the dual problem. We denote by $\tilde{G}_\parallel^\diamond$ the choice of $\tilde{G}_\parallel \in \mathcal{S}$ that minimizes Eq. (36), and we define

$$\tilde{G}^\diamond := G_\perp - \tilde{G}_\parallel^\diamond. \tag{37}$$

Then $\tilde{G}^*$ that maximizes Eq. (34) has the form

$$\tilde{G}^* = \tilde{\rho}_0^\diamond - \tilde{\rho}_1^\diamond, \tag{38}$$

where $\tilde{\rho}_0^\diamond$ is a density operator supported on the eigenspace of $\tilde{G}^\diamond$ with the maximal eigenvalue, and $\tilde{\rho}_1^\diamond$ is a density operator supported on the eigenspace of $\tilde{G}^\diamond$ with the minimal eigenvalue. The minimization in Eq. (36) ensures that $\tilde{G}^*$ of this form can be chosen to obey the constraint Eq. (33).

In the noiseless case ($\mathcal{S} = \text{span}\{I\}$), the minimum in Eq. (36) occurs when the maximum and minimum eigenvalues $G_\perp - \tilde{G}_\parallel$ have the same absolute value, and then the operator norm is half the difference of the maximum and minimum eigenvalues of $G_\perp$. Hence, we recover the result Eq. (23). When noise is introduced, $\mathcal{S}$ swells and the minimal distance shrinks, lowering the QFI and reducing the precision of parameter estimation. If HNLS fails, then the minimum distance is zero, and no QEC code can achieve HL scaling, in accord with Theorem 1.

**Kerr effect with photon loss.** To illustrate how the optimization procedure works, consider a bosonic mode with the nonlinear (Kerr effect[50]) Hamiltonian

$$H(\omega) = \omega \left(a^\dagger a\right)^2, \tag{39}$$

where our objective is to estimate $\omega$. In this case, the probe is infinite dimensional, but suppose we assume that the occupation number $n = a^\dagger a$ is bounded: $n \leq \overline{n}$, where $\overline{n}$ is even. The noise source is photon loss, with Lindblad jump operator $L \propto a$. Can we find a QEC code that protects the probe against loss and achieves HL scaling for estimation of $\omega$?

To solve the dual program, we find real parameters $\alpha, \beta, \gamma, \delta$, which minimize the operator norm of

$$\tilde{n}^2 := n^2 + \alpha n + \beta a + \gamma a^\dagger + \delta, \tag{40}$$

where $n \leq \overline{n}$. Since $a$ and $a^\dagger$ are off-diagonal in the occupation number basis, we should set $\beta$ and $\gamma$ to zero for the purpose of minimizing the difference between the largest and smallest eigenvalue of $\tilde{n}^2$. After choosing $\alpha$ such that $\tilde{n}^2$ is minimized at $n = \overline{n}/2$, and choosing $\delta$ so that the maximum and minimum eigenvalues of $\tilde{n}^2$ are equal in absolute value and opposite in sign, we have

$$\left(\tilde{n}^2\right)^\diamond = \left(n - \frac{1}{2}\overline{n}\right)^2 - \frac{1}{8}\overline{n}^2, \tag{41}$$

which has operator norm $\left\| \left(\tilde{n}^2\right)^\diamond \right\| = \overline{n}^2/8$; hence the optimal QFI after evolution time $t$ is $\mathcal{F}(\rho(t)) = t^2\overline{n}^4/16$, according to Eq. (36). For comparison, the minimal operator norm is $\overline{n}^2/2$ for a noiseless bosonic mode with $n \leq \overline{n}$. We see that loss reduces

the precision of our estimate of $\omega$, but only by a factor of 4 if we use the optimal QEC code. HL scaling can still be maintained. The scaling $\delta\hat{\omega} \sim 1/\overline{n}^2$ of the optimal precision arises from the nonlinear boson-boson interactions in the Hamiltonian Eq. (39)[51].

To find the code states, we note that the eigenstate of $(\tilde{n}^2)^\diamond$ with the lowest eigenvalue $-\overline{n}^2/8$ is $|n = \overline{n}/2\rangle$, while the largest eigenvalue $+\overline{n}^2/8$ has the two degenerate eigenstates $|n = 0\rangle$ and $|n = \overline{n}\rangle$. The code condition (2) requires that both code vectors have the same expectation value of $L^\dagger L \propto n$, and we therefore may choose

$$|C_0\rangle = |\overline{n}/2\rangle_P \otimes |0\rangle_A, \quad |C_1\rangle = \frac{1}{\sqrt{2}}\left(|0\rangle_P + |\overline{n}\rangle_P\right) \otimes |1\rangle_A \tag{42}$$

as the code achieving optimal precision. For $\overline{n} \geq 4$, the ancilla may be discarded, and we can use the simpler code

$$|C_0\rangle = |\overline{n}/2\rangle_P, \quad |C_1\rangle = \frac{1}{\sqrt{2}}\left(|0\rangle_P + |\overline{n}\rangle_P\right), \tag{43}$$

which is easier to realize experimentally. Eqs. (17) and (18) are still satisfied without the ancilla, because the states $\{|C_0\rangle, |C_1\rangle, a|C_0\rangle, a|C_1\rangle\}$ are all mutually orthogonal. This encoding Eq. (43) belongs to the family of "binomial quantum codes" which, as discussed in ref. [52], can protect against loss of bosonic excitations.

An experimental realization of this coding scheme can be achieved using tools from circuit quantum electrodynamics, by coupling a single transmon qubit to two microwave waveguide resonators. For example, when $\overline{n}$ is a multiple of 4, $|C_0\rangle$ and $|C_1\rangle$ both have even photon parity while $a|C_0\rangle$ and $a|C_1\rangle$ both have odd parity. Then QEC can be carried out by the following procedure: (1) a quantum non-demolition parity measurement is performed to check whether photon loss has occurred[38,53]. (2) If photon loss is detected, the initial logical encoding is restored using optimal control pulses[38,39]. (3) If there is no photon loss, the quantum state is projected onto the code space $\text{span}\{|C_0\rangle, |C_1\rangle\}$[54]. The probability of an uncorrectable logical error becomes arbitrarily small if the QEC procedure is sufficiently fast compared to the photon loss rate. Meanwhile, the Kerr signal accumulates coherently in the relative phase of $|C_0\rangle$ and $|C_1\rangle$, so that HL scaling can be attained for arbitrarily fast quantum control. For integer values of $\overline{n}$ that are not a multiple of 4, coding schemes can still be constructed that protect against photon loss, as described in ref. [52].

**Approximate error correction.** Generic Markovian noise is full rank, which means that the span $\mathcal{S}$ is the full Hilbert space $\mathcal{H}_P$ of the probe; hence the HNLS criterion of Theorem 1 is violated for any probe Hamiltonian $H(\omega)$, and asymptotic SQL scaling cannot be surpassed. Therefore, for any Markovian noise model that meets the HNLS criterion, the HL scaling achieved by our QEC code is not robust against generic small perturbations of the noise model.

We should therefore emphasize that a substantial improvement in precision can be achieved using a QEC code even in cases where HNLS is violated. Consider in particular a Markovian master equation with Lindblad operators divided into two sets $\{L_k\}$ (L-type noise) and $\{J_m\}$ (J-type noise), where the J-type noise is parametrically weak, with noise strength

$$\epsilon := \left\| \sum_m J_m^\dagger J_m \right\| \tag{44}$$

($\|\cdot\|$ denotes the operator norm). If we use the optimal code that protects against L-type noise, then the joint logical state of

probe and ancilla evolves according to a modified master equation, with Hamiltonian $H_{eff} = \Pi_C H \Pi_C$, and effective Lindblad operators $J_{m,j}$ acting within the code space, where

$$\left\| \sum_{m,j} J_{m,j}^\dagger J_{m,j} \right\| \le \epsilon. \qquad (45)$$

(See the Methods for further discussion.)

This means that the state of the error-corrected probe deviates by a distance $O(\epsilon t)$ (in the $L^1$ norm) from the (effectively noiseless) evolution in the absence of $J$-type noise. Therefore, using this code, the QFI of the error-corrected probe increases quadratically in time (and the precision $\delta\hat{\omega}$ scales like $1/t$) up until an evolution time $t \propto 1/\epsilon$, before crossing over to asymptotic SQL scaling.

## Discussion

Noise limits the precision of quantum sensing. Quantum error correction can suppress the damaging effects of noise, thereby improving the fidelity of quantum information processing and quantum communication, but whether QEC improves the efficacy of quantum sensing depends on the structure of the noise and the signal Hamiltonian. Unless suitable conditions are met, the QEC code that tames the noise might obscure the signal as well, nullifying the advantages of QEC.

Our study of quantum sensing using a noisy probe has focused on whether the precision $\delta$ of parameter estimation scales asymptotically with the total sensing time $t$ as $\delta \propto 1/t$ (HL) or $\delta \propto 1/\sqrt{t}$ (SQL). We have investigated this question in an idealized setting, where the experimentalist has access to noiseless (or correctable) ancillas and can apply quantum controls that are arbitrarily fast and accurate, and we have also assumed that the noise acting on the probe is Markovian. Under these assumptions, we have found the general criterion for HL scaling to be achievable, the HNLS criterion. If HNLS is satisfied, a QEC code can be constructed that achieves HL scaling, and if HNLS is violated, then SQL scaling cannot be surpassed.

In the case where HNLS is satisfied, we have seen that the QEC code achieving the optimal precision can be chosen to be two-dimensional, and we have described an algorithm for constructing this optimal code. The precision attained by this code has a geometrical interpretation in terms of the minimal distance (in the operator norm) of the signal Hamiltonian from the "Lindblad span" $\mathcal{S}$, the subspace spanned by $I$, $L_k$, $L_k^\dagger$, and $L_k^\dagger L_j$, where $\{L_k\}$ is the set of Lindblad jump operators appearing in the probe's Markovian master equation.

Many questions merit further investigation. We have focused on the dichotomy of HL vs. SQL scaling, but it is also worthwhile to characterize constant factor improvements in precision that can be achieved using QEC in cases where HNLS is violated[55]. We should clarify the applications of QEC to sensing when quantum controls have realistic accuracy and speed. Finally, it is interesting to consider probes subject to non-Markovian noise. In that case, tools such as dynamical decoupling[56–59] can mitigate noise, but just as for QEC, we need to balance desirable suppression of the noise against undesirable suppression of the signal in order to formulate the most effective sensing strategy.

*Note added:* During the preparation of this manuscript, we became aware of related work by Demkowicz-Dobrzański et al.[60], which provided a similar proof of the necessary condition in Theorem 1 and an equivalent description of the QEC conditions Eqs. (17), (18), and (20). We and the authors of ref. [60] obtained this result independently. Both our paper and ref. [60] generalize results obtained earlier in ref. [37].

## Methods

**Linear scaling of the QFI.** Here we prove that the QFI scales linearly with the evolution time $t$ in the case where the HNLS condition is violated. We follow the proof in ref. [37], which applies when the probe is a qubit, and generalize their proof to the case where the probe is $d$-dimensional.

We approximate the quantum channel

$$\mathcal{E}_{dt}(\rho) = \rho - i\omega[G, \rho]dt$$
$$+ \sum_{k=1}^{r} \left( L_k \rho L_k^\dagger - \tfrac{1}{2}\left\{ L_k^\dagger L_k, \rho \right\} \right) dt + O(dt^2) \qquad (46)$$

by the following one:

$$\tilde{\mathcal{E}}_{dt}(\rho) = \sum_{k=0}^{r} K_k \rho K_k^\dagger, \qquad (47)$$

where $K_0 = I + \left( -i\omega G - \tfrac{1}{2}\sum_{k=1}^{r} L_k^\dagger L_k \right) dt$ and $K_k = L_k \sqrt{dt}$ for $k \ge 1$. The approximation is valid because the distance between $\mathcal{E}_{dt}$ and $\tilde{\mathcal{E}}_{dt}$ is $O(dt^2)$ and the sensing time is divided into $\frac{t}{dt}$ segments, meaning the error $O\left(\frac{t}{dt} \cdot dt^2\right) = O(tdt)$ introduced by this approximation in calculating the QFI vanishes as $dt \to 0$. Next, we calculate the operators $\alpha_{dt} = \left(\dot{\mathbf{K}} - ih\mathbf{K}\right)^\dagger \left(\dot{\mathbf{K}} - ih\mathbf{K}\right)$ and $\beta_{dt} = i\left(\dot{\mathbf{K}} - ih\mathbf{K}\right)^\dagger \mathbf{K}$ for the channel $\mathcal{E}_{dt}(\rho)$, and expand these operators as a power series in $\sqrt{dt}$:

$$\alpha_{dt} = \alpha^{(0)} + \alpha^{(1)}\sqrt{dt} + \alpha^{(2)}dt + O\left(dt^{3/2}\right), \qquad (48)$$

$$\beta_{dt} = \beta^{(0)} + \beta^{(1)}\sqrt{dt} + \beta^{(2)}dt + \beta^{(3)}dt^{3/2} + O\left(dt^2\right). \qquad (49)$$

We will now search for a Hermitian matrix $h$ that sets low-order terms in each power series to 0.

Expanding $h$ as $h = h^{(0)} + h^{(1)}\sqrt{dt} + h^{(2)}dt + h^{(3)}dt^{3/2} + O(dt^2)$ in $\sqrt{dt}$, and using the notation $(K_0, K_1, \ldots, K_r)^T = \mathbf{K} = \mathbf{K}^{(0)} + \mathbf{K}^{(1)}dt^{1/2} + \mathbf{K}^{(2)}dt$, we find

$$\alpha^{(0)} = \mathbf{K}^{(0)\dagger} h^{(0)} h^{(0)} \mathbf{K}^{(0)} = \sum_{k=0}^{r} \left| h_{0k}^{(0)} \right|^2 I = 0$$
$$\Rightarrow h_{0k}^{(0)} = 0, \ 0 \le k \le r. \qquad (50)$$

Therefore $h^{(0)}\mathbf{K}^{(0)} = \mathbf{0}$ and $\alpha^{(1)} = \beta^{(0)} = 0$ are automatically satisfied. Then,

$$\beta^{(1)} = -\mathbf{K}^{(0)\dagger} h^{(1)} \mathbf{K}^{(0)} = -h_{00}^{(1)} I = 0 \Rightarrow h_{00}^{(1)} = 0. \qquad (51)$$

and

$$\beta^{(2)} = i\dot{\mathbf{K}}^{(2)\dagger}\mathbf{K}^{(0)} - \mathbf{K}^{(1)\dagger} h^{(0)}\mathbf{K}^{(1)}$$
$$- \mathbf{K}^{(0)\dagger} h^{(1)}\mathbf{K}^{(1)} - \mathbf{K}^{(1)\dagger} h^{(1)}\mathbf{K}^{(0)} - \mathbf{K}^{(0)\dagger} h^{(2)}\mathbf{K}^{(0)}$$
$$= G - \sum_{k,j=1}^{r} h_{jk}^{(0)} L_k^\dagger L_j - \sum_{k=1}^{r} \left( h_{0k}^{(1)} L_k + h_{k0}^{(1)} L_k^\dagger \right) - h_{00}^{(2)} I, \qquad (52)$$

which can be set to 0 if and only if $G$ is a linear combination of $I$, $L_k$, $L_k^\dagger$ and $L_k^\dagger L_j$ ($0 \le k, j \le r$).

In addition,

$$\beta^{(3)} = -\mathbf{K}^{(1)\dagger} h^{(1)}\mathbf{K}^{(1)} - \mathbf{K}^{(0)\dagger} h^{(2)}\mathbf{K}^{(1)}$$
$$- \mathbf{K}^{(1)\dagger} h^{(2)}\mathbf{K}^{(0)} - \mathbf{K}^{(0)\dagger} h^{(3)}\mathbf{K}^{(0)}$$
$$= -\sum_{k,j=1}^{r} h_{jk}^{(1)} L_k^\dagger L_j - \sum_{k=1}^{r} \left( h_{0k}^{(2)} L_k + h_{k0}^{(2)} L_k^\dagger \right) - h_{00}^{(3)} I = 0 \qquad (53)$$

can be satisfied by setting the above parameters (which do not appear in the expressions for $\alpha^{(0,1)}$ and $\beta^{(0,1,2)}$) all to 0 (other terms in $\beta^{(3)}$ are 0 because of the constraints on $h^{(0)}$ and $h^{(1)}$ in Eqs. (50) and (51)). Therefore, when $G$ is a linear combination of $I$, $L_k$, $L_k^\dagger$ and $L_k^\dagger L_j$, there exists an $h$ such that $\alpha_{dt} = O(dt)$ and $\beta_{dt} = O(dt^2)$ for the quantum channel $\mathcal{E}_{dt}$; therefore the QFI obeys

$$\mathcal{F}(\rho(t)) \le 4\tfrac{t}{dt}\|\alpha_{dt}\| + 4\left(\tfrac{t}{dt}\right)^2 \|\beta_{dt}\| \left( \|\beta_{dt}\| + 2\sqrt{\|\alpha_{dt}\|} \right)$$
$$= 4\|\alpha^{(2)}\|t + O(\sqrt{dt}), \qquad (54)$$

in which $\alpha^{(2)} = \left( h^{(1)}\mathbf{K}^{(0)} + h^{(0)}\mathbf{K}^{(1)} \right)^\dagger \left( h^{(1)}\mathbf{K}^{(0)} + h^{(0)}\mathbf{K}^{(1)} \right)$ under the constraint $\beta^{(2)} = 0$.

**The QEC condition.** Here we consider the quantum channel Eq. (2), which describes the joint evolution of a noisy quantum probe and noiseless ancilla over time interval $dt$. Suppose that a QEC code obeys the conditions (1) and (2) in Eqs. (17) and (18), where $\Pi_C$ is the orthogonal projector onto the code space. We will construct a recovery operator such that the error-corrected time evolution is unitary to linear order in $dt$, governed by the effective Hamiltonian $H_{eff} = \omega\Pi_C G\Pi_C$.

For a density operator $\rho = \Pi_C \rho \Pi_C$ in the code space, conditions (1) and (2) imply

$$\Pi_C \mathcal{E}_{dt}(\rho) \Pi_C = \rho - i\omega[\Pi_C G \Pi_C, \rho]dt + \sum_{k=1}^{r} (|\lambda_k|^2 - \mu_{kk})\rho dt + O(dt^2), \tag{55}$$

$$\Pi_E \mathcal{E}_{dt}(\rho) \Pi_E = \sum_{k=1}^{r} (L_k - \lambda_k)\rho(L_k^\dagger - \lambda_k^*)dt + O(dt^2), \tag{56}$$

where $\Pi_E = I - \Pi_C$. When acting on a state in the code space, $\Pi_E \mathcal{E}_d t(\cdot)\Pi_E$ is an operation with Kraus operators

$$K_k = (I - \Pi_C) L_k \Pi_C \sqrt{dt}, \tag{57}$$

which obey the normalization condition

$$\sum_{k=1}^{r} K_k^\dagger K_k = \sum_{k=1}^{r} \Pi_C L_k^\dagger (I - \Pi_C) L_k \Pi_C dt = \sum_{k=1}^{r} (\mu_{kk} - |\lambda_k|^2)dt, \tag{58}$$

where we have used conditions (1) and (2). Therefore, if $\rho$ is in the code space, then a recovery channel $\mathcal{R}_E(\cdot)$ such that

$$\mathcal{R}_E(\Pi_E \mathcal{E}_{dt}(\rho)\Pi_E) = -\sum_{k=1}^{r} (|\lambda_k|^2 - \mu_{kk})\rho dt + O(dt^2) \tag{59}$$

can be constructed, provided that the operators $\{L_k - \lambda_k\}_{k=1}^{r}$ satisfy the standard QEC conditions[31–33]. Indeed, these conditions are satisfied because $\Pi_C \left(L_k^\dagger - \lambda_k^*\right)\left(L_j - \lambda_j\right)\Pi_C = \left(\mu_{kj} - \lambda_k^* \lambda_j\right)\Pi_C$, for all $k, j$. Therefore, the quantum channel

$$\mathcal{R}(\sigma) = \Pi_C \sigma \Pi_C + \mathcal{R}_E(\Pi_E \sigma \Pi_E) \tag{60}$$

completely reverses the effects of the noise. The channel describing time evolution for time $dt$ followed by an instantaneous recovery step is

$$\mathcal{R}(\mathcal{E}_{dt}(\rho)) = \rho - i\omega[\Pi_C G \Pi_C, \rho]dt + O(dt^2), \tag{61}$$

a noiseless unitary channel with effective Hamiltonian $\omega \Pi_C G \Pi_C$ if $O(dt^2)$ corrections are neglected.

The dependence of the Hamiltonian on $\omega$ can be detected, for a suitable initial code state $\rho_{\text{in}}$, if and only if $\Pi_C G \Pi_C$ has at least two distinct eigenvalues. Thus, for nontrivial error-corrected sensing we require condition (3): $\Pi_C G \Pi_C \neq \text{constant} \times \Pi_C$.

**Error-correctable noisy ancillas.** In the main text, we assumed that a noiseless ancilla system is available for the purpose of constructing the QEC code. Here, we relax that assumption. We suppose instead that the ancilla is subject to Markovian noise, which is uncorrelated with noise acting on the probe. Hence, the joint evolution of probe and ancilla during the infinitesimal time interval $dt$ is described by the quantum channel

$$\mathcal{E}_{dt}(\rho) = \rho - i\omega[G \otimes I, \rho]dt + \sum_{k=1}^{r} \left((L_k \otimes I)\rho(L_k^\dagger \otimes I) - \frac{1}{2}\{L_k^\dagger L_k \otimes I, \rho\}\right)dt + \sum_{k'=1}^{r'} \left((I \otimes L_k')\rho(I \otimes L_k'^\dagger) - \frac{1}{2}\{I \otimes L_k'^\dagger L_k', \rho\}\right)dt + O(dt^2), \tag{62}$$

where $\{L_k\}$ are Lindblad jump operators acting on the probe, and $\{L_k'\}$ are Lindblad jump operators acting on the ancilla.

In this case, we may be able to protect the probe using a code $\overline{C}$ scheme with two layers—an "inner code" $C'$ and an "outer code" $C$. Assuming as before that arbitrarily fast and accurate quantum processing can be performed, and that the Markovian noise acting on the ancilla obeys a suitable condition, an effectively noiseless encoded ancilla can be constructed using the inner code. Then, the QEC scheme that achieves HL scaling can be constructed using the same method as in the main text, but with the encoded ancilla now playing the role of the noiseless ancilla used in our previous construction.

Errors on the ancilla can be corrected if the projector $\Pi_{C'}$ onto the inner code $C'$ satisfies the conditions.

$$[1'] \ \Pi_{C'} L_k' \Pi_{C'} = \lambda_k' \Pi_{C'}, \ \forall k, \tag{63}$$

$$[2'] \ \Pi_{C'} L_j'^\dagger L_k' \Pi_{C'} = \mu_{jk}' \Pi_{C'}, \ \forall k, j. \tag{64}$$

Eqs. (63) and (64) resemble Eqs. (17) and (18), except that the inner code $C'$ is supported only on the ancilla system $\mathcal{H}_A$, while the code $C$ in Eqs. (17) and (18) is

supported on the joint system $\mathcal{H}_P \otimes \mathcal{H}_A$ of probe and ancilla. To search for a suitable inner code $C'$, we may use standard QEC methods; namely we seek an encoding of the logical ancilla with sufficient redundancy for Eqs. (63) and (64) to be satisfied.

Given a code $C$ that satisfies Eqs. (17), (18), and (20) for the case of a noiseless ancilla, and a code $C'$ supported on a noisy ancilla that satisfies Eqs. (63) and (64), we construct the code $\overline{C}$ that achieves HL scaling for a noisy ancilla system by "concatenating" the inner code $C'$ and the outer code $C$. That is, if the basis states for the code $C$ are $\{|C_0\rangle, |C_1\rangle\}$, where

$$|C_i\rangle = \sum_{j,k=1}^{d} C_i^{(jk)} |j\rangle_P \otimes |k\rangle_A, \tag{65}$$

then the corresponding basis states for the code $\overline{C}$ are $|\overline{C}_0\rangle, |\overline{C}_1\rangle$, where

$$|\overline{C}_i\rangle = \sum_{j,k=1}^{d} C_i^{(jk)} |j\rangle_P \otimes |C_k'\rangle_A, \tag{66}$$

and $|C_k'\rangle$ denotes the basis state of $C'$ which encodes $|k\rangle$. Using our fast quantum controls, the code $C'$ protects the ancilla against the Markovian noise, and the code $\overline{C}$ then protects the probe, so that HL scaling is achievable.

In fact, the code that achieves HL scaling need not have this concatenated structure; any code that corrects both the noise acting on the probe and the noise acting on the ancilla will do. For Markovian noise acting independently on probe and ancilla as in Eq. (62), the conditions Eqs. (17) and (18) on the QEC code should be generalized to

$$\Pi_{\overline{C}}(O \otimes O')\Pi_{\overline{C}} \propto \Pi_{\overline{C}}, \quad \forall O \in \mathcal{S} \text{ and } O' \in \mathcal{S}'; \tag{67}$$

here $\mathcal{S} = \text{span}\{I, L_k, L_k^\dagger, L_j^\dagger L_k, \forall k, j\}$, $\mathcal{S}' = \text{span}\{I, L_k', L_k'^\dagger, L_j'^\dagger L_k', \forall k, j\}$, and $\Pi_{\overline{C}}$ is the projector onto the code $\overline{C}$ supported on $\mathcal{H}_P \otimes \mathcal{H}_A$. The condition Eq. (20) remains the same as before, but now applied to the code $\overline{C}$: $\Pi_{\overline{C}}(G \otimes I)\Pi_{\overline{C}} \neq \text{constant } \Pi_{\overline{C}}$. When these conditions are satisfied, the noise acting on probe and ancilla is correctable; rapidly applying QEC makes the evolution of the probe effectively unitary (and nontrivial), to linear order in $dt$.

**Robustness of the QEC scheme.** We consider the following quantum channel, where the "$J$-type noise," with Lindblad operators $\{J_m\}_{m=1}^{r_2}$, is regarded as a small perturbation:

$$\mathcal{E}_{dt}(\rho) = \rho - i\omega[G, \rho]dt + \sum_{k=1}^{r_1} \left(L_k \rho L_k^\dagger - \frac{1}{2}\{L_k^\dagger L_k, \rho\}\right)dt + \sum_{m=1}^{r_2} \left(J_m \rho J_m^\dagger - \frac{1}{2}\{J_m^\dagger J_m, \rho\}\right)dt + O(dt^2). \tag{68}$$

We assume that the "$L$-type noise," with Lindblad operators $\{L_k\}_{k=1}^{r_1}$, obeys the QEC conditions (1) and (2), and that $\mathcal{R}$ is the recovery operation that corrects this noise. By applying this recovery step after the action of $\mathcal{E}_{dt}$ on a state $\rho$ in the code space, we obtain a modified channel with residual $J$-type noise.

Suppose that $\mathcal{R}$ has the Kraus operator decomposition $\mathcal{R}(\sigma) = \sum_{j=1}^{s} R_j \sigma R_j^\dagger$, where $\sum_{j=1}^{s} R_j^\dagger R_j = I$. We also assume that $R_j = \Pi_C R_j$, because the recovery procedure has been constructed such that the state after recovery is always in the code space. Then

$$\mathcal{R}(\mathcal{E}_{dt}(\rho)) = \rho - i\omega[\Pi_C G \Pi_C, \rho]dt + \sum_{m=1}^{r_2}\sum_{j=1}^{s} \left(J_{m,j}^{(C)} \rho J_{m,j}^{(C)\dagger} - \frac{1}{2}\{J_{m,j}^{(C)\dagger} J_{m,j}^{(C)}, \rho\}\right)dt + O(dt^2), \tag{69}$$

where $\left\{J_{m,j}^{(C)} = \Pi_C R_j J_m \Pi_C\right\}$ are the effective Lindblad operators acting on code states.

The trace ($L^1$) distance[31] between the unitarily evolving state Eq. (61) and the state subjected to the residual noise Eq. (69) is bounded above by

$$\frac{1}{2}\max_\rho \text{tr}\left|\sum_{m,j} J_{m,j}^{(C)} \rho J_{m,j}^{(C)\dagger}\right|dt$$
$$+ \frac{1}{4}\max_\rho \text{tr}\left|\sum_{m,j} J_{m,j}^{(C)\dagger} J_{m,j}^{(C)} \rho + \rho \sum_{m,j} J_{m,j}^{(C)\dagger} J_{m,j}^{(C)}\right|dt$$
$$\leq \left\|\sum_{m=1}^{r_2}\sum_{j=1}^{s} J_{m,j}^{(C)\dagger} J_{m,j}^{(C)}\right\|dt = \left\|\Pi_C\left(\sum_{m=1}^{r_2} J_m^\dagger J_m\right)\Pi_C\right\|dt$$
$$\leq \left\|\sum_{m=1}^{r_2} J_m^\dagger J_m\right\|dt \tag{70}$$

to first order in $dt$, where $\|\cdot\|$ denotes the operator norm. If the noise strength

$$\epsilon := \left\|\sum_{m=1}^{r_2} J_m^\dagger J_m\right\| \tag{71}$$

of the Lindblad operators $\{J_m\}_{m=1}^{r_2}$ is low, the evolution is approximately unitary when $t \ll 1/\epsilon$. In this sense, the QEC scheme is robust against small $J$-type noise.

**Code optimization as a semidefinite program**. Optimization of the QEC code can be formulated as the following optimization problem:

$$\text{maximize} \quad \text{tr}(\tilde{G}G_\perp)$$
$$\text{subject to} \; \text{tr}(|\tilde{G}|) \leq 2 \text{ and } \text{tr}(\tilde{G}) = \text{tr}(\tilde{G}L_k) \tag{72}$$
$$= \text{tr}(\tilde{G}L_k^\dagger L_j) = 0, \quad \forall j, k.$$

This optimization problem is convex (because $\text{tr}|\cdot|$ is convex) and satisfies the Slater's condition, so it can be solved by solving its Lagrange dual problem[61]. The Lagrangian $L(\tilde{G}, \lambda, \nu)$ is defined for $\lambda \geq 0$ and $\nu_k \in \mathbb{R}$:

$$L(\tilde{G}, \lambda, \nu) = \text{tr}(\tilde{G}G_\perp) - \lambda(\text{tr}(|\tilde{G}|) - 2) + \sum_k \nu_k \text{tr}(E_k \tilde{G}), \tag{73}$$

where $\{E_k\}$ is any basis of $\mathcal{S}$. The optimal value is obtained by taking the minimum of the dual

$$
\begin{aligned}
g(\lambda, \nu) &= \max_{\tilde{G}} L(\tilde{G}, \lambda, \nu) \\
&= \max_{\tilde{G}} \text{tr}\left(\left(G_\perp + \sum_k \nu_k E_k\right)\tilde{G} - \lambda|\tilde{G}|\right) + 2\lambda \\
&= \begin{cases} 2\lambda & \lambda \geq \left\|G_\perp + \sum_k \nu_k E_k\right\| \\ \infty & \lambda \leq \left\|G_\perp + \sum_k \nu_k E_k\right\| \end{cases}
\end{aligned} \tag{74}
$$

over $\lambda$ and $\{\nu_k\}$, where $\|\cdot\| = \max_{|\psi\rangle} |\langle\psi| \cdot |\psi\rangle|$ is the operator norm. Hence the optimal value of the primal problem is

$$\min_{\lambda, \nu} g(\lambda, \nu) = 2\min_{\nu_k} \left\|G_\perp + \sum_k \nu_k E_k\right\| = 2\min_{\tilde{G}_\parallel \in \mathcal{S}} \|G_\perp - \tilde{G}_\parallel\|. \tag{75}$$

The optimization problem Eq. (75) is equivalent to the following SDP:[61]

$$\text{minimize} \; s$$
$$\text{subject to} \begin{pmatrix} sI & G_\perp + \sum_k \nu_k E_k \\ G_\perp + \sum_k \nu_k E_k & sI \end{pmatrix} \succeq 0 \tag{76}$$

for variables $\nu_k \in \mathbb{R}$ and $s \succeq 0$. Here "$\succeq 0$" denotes positive semidefinite matrices. SDPs can be solved using the Matlab-based package CVX[62].

Once we have the solution to the dual problem, we can use it to find the solution to the primal problem. We denote by $\lambda^\diamond$ and $\nu^\diamond$ the values of $\lambda$ and $\nu$ where $g(\lambda, \nu)$ attains its minimum, and define

$$\tilde{G}^\diamond = G_\perp + \sum_k \nu_k^\diamond E_k. \tag{77}$$

The minimum $g(\lambda^\diamond, \nu^\diamond)$ matches the value of the Lagrangian $L(\tilde{G}, \lambda^\diamond, \nu^\diamond)$ when $\tilde{G} = \tilde{G}^*$ is the value of $\tilde{G}$ that maximizes $\text{tr}(\tilde{G}G_\perp)$ subject to the constraints. This means that

$$\text{tr}(\tilde{G}^*\tilde{G}^\diamond) = 2\|\tilde{G}^\diamond\|. \tag{78}$$

Since we require $\text{tr}(\tilde{G}^*) = 0$ and $\text{tr}|\tilde{G}^*| = 2$, and because minimizing $g(\lambda, \nu)$ enforces that the maximum and minimal eigenvalues of $\tilde{G}^\diamond$ have the same absolute value and opposite sign, we conclude that

$$\tilde{G}^* = \tilde{\rho}_0^\diamond - \tilde{\rho}_1^\diamond, \tag{79}$$

where $\tilde{\rho}_0^\diamond$ is a density operator supported on the eigenspace of $\tilde{G}^\diamond$ with the maximal eigenvalue, and $\tilde{\rho}_1^\diamond$ is a density operator supported on the eigenspace of $\tilde{G}^\diamond$ with the minimal eigenvalue. A $\tilde{G}^*$ of this form which satisfies the constraints of the primal problem is guaranteed to exist.

**Data availability**. Data sharing not applicable to this article as no data sets were generated or analyzed during the current study.

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

## Acknowledgements

We thank Fernando Brandão, Yanbei Chen, Steve Girvin, Linshu Li, Mikhail Lukin, Changling Zou for inspiring discussions. We acknowledge support from the ARL-CDQI (W911NF-15-2-0067), ARO (W911NF-14-1-0011, W911NF-14-1-0563), ARO MURI (W911NF-16-1-0349), AFOSR MURI (FA9550-14-1-0052, FA9550-15-1-0015), NSF (EFMA-1640959), Alfred P. Sloan Foundation (BR2013-049), and Packard Foundation (2013-39273). The Institute for Quantum Information and Matter is an NSF Physics Frontiers Center with support from the Gordon and Betty Moore Foundation.

## Author contributions

J.P. and L.J. conceived this project. S.Z. proved linear scaling of the QFI and constructed the QEC code. S.Z., M.Z. and L.J. formulated the QEC condition. S.Z., J.P. and L.J. wrote the manuscript.

## Additional information

**Competing interests:** The authors declare no competing financial interests.

