## [Peer Review File · Nature Communications]

Reviewers' comments:

Reviewer #1 (Remarks to the Author):

I find this to be an excellent paper which provides an elegant and novel solution to an outstanding problem in quantum metrology.

Exploiting quantum effects to improve high-precision sensing holds great promise, with applications ranging from magnetometry for medical diagnostics to optical interferometry for gravitational wave detection. Under ideal conditions, quantum metrology can achieve a quadratic improvement in resource scaling over classical methods (from standard quantum limited to Heisenberg limited scaling), leading to spectacular precision gains for large probes or extended measurement times. However, it is crucial to understand how quantum-enhanced methods perform in the presence of realistic imperfections and noise. In recent years, this research direction has attracted much effort, and significant progress has been made.

A general framework for addressing and fighting the effects of noise is quantum error correction (QEC). Previous works had shown that noise tends to negate any scaling advantage for quantum metrology, but that QEC can indeed protect the quantum metrological gains for some particular noise types. In this paper, the authors provide a necessary and sufficient condition for QEC to enable Heisenberg limited precision scaling under general Markovian noise. In addition, they identify an error correcting code achieving this scaling, and show how to optimise it using semi-definite programming.

This is a strong theoretical result, which answers the question of when QEC is helpful for quantum metrology for a broad class of noise. It will surely influence future theory work on quantum metrology, and is likely to inform experimental efforts, e.g. in quantum magnetometry. I note that while the work Ref. [43] also proves the necessity of the condition in the present manuscript, it does not address sufficiency, and the constructive proof with a concrete error correcting code provided here is elegant.

I have a few minor comments, which I would recommend the authors address before publication:

1)

On p.2 below Fig. 1, the authors mention previous work showing that HL scaling can be achieved for qubits with perpendicular noise using QEC. They cite Ref. [16] (Escher et al), but to my knowledge this work did not address QEC. Perhaps the authors meant to cite Ref. [30] here?

2)

In the same paragraph, the σ 's are introduced but never defined to be the Pauli operators.

3)

Below eq. (2), the authors refer to the number of jump operators in the master equation as the noise 'rank'. However, it is unclear to me whether this is consistent with terminology in other works. For example, in Ref. [30], it is the rank of the matrix appearing in the master equation dissipator (in the Pauli basis). Could the authors clarify whether this terminology is standard?

4)

In the last paragraph of p. 2 and top of p.3, the authors argue that when the Hamiltonian is nonlinear in the estimated parameter, one can first obtain sufficient precision on the parameter such that estimation is effectively local and the dependence can be treated as linear, without affecting the asymptotic scaling. To my knowledge, the quantum Cramer-Rao bound eq. (1) can only be ensured to be tight for local estimation, so one would anyway need to work in this regime.

5)

In the first paragraph below Thm. 1, the authors consider a qubit example with a Lindblad operator given by a complex linear combination of Pauli operators, and conclude that the real and imaginary parts of the vector defining the noise must be parallel for it to be correctible. This was also derived in Ref. [30], so a citation here would be in order.

6)

A typo: 5 lines below eq. (6) 'correcting' should be 'corrected'.

7)

The λ and μ appearing in eqs (16) and (17) are never defined.

8)

Below eq. (19), some more explanation of why eqs. (18) and (19) imply that the conditions in eq. (15) are satisfied would be good.

9)

Below eq. (22), the authors say that the Cramer-Rao bound can be saturated. I think this is true only asymptotically for local estimation.

10)

Below eq. (35), the authors find that the quantum Fisher information scales with the photon number to the fourth power. This seems to me to be due to the nonlinearity of the Hamiltonian, which induces photon-photon interactions (it is quadratic in the photon number). The scaling is consistent with that found for quantum metrology with interactions in Phys. Rev. Lett. 98, 090401 (2007). The authors might want to comment on this connection.

Reviewer #2 (Remarks to the Author):

The submitted work “Adaptive quantum metrology under general Markovian noise” seems to resolve one of the most troublesome problems in the field of quantum metrology. In order to fully understand its significance, one has to consider a brief history of the field throughout last years.

The main message of quantum metrology at its early stage was that the precision of estimation of an unknown parameter acquires a quadratic gain in the size of a resource (number of probed systems, or total probing time) when quantum resources (depending on the scheme: entanglement or quantum coherence) are used. The ultimate limit of the precision of estimation in quantum-enhanced metrology is commonly known as the Heisenberg Limit (HL), whereas the ultimate classical limit of precision is (a bit confusingly) called Standard Quantum Limit (SQL). It was early recognized that under noisy conditions achieving the HL could not be possible, which was most concisely presented in the ref. [17]. Namely, it was shown that even arbitrarily small full-rank noise forces a classical (SQL) asymptotic scaling of precision. This result was treated by many from among the metrological community as a final stroke against the entire field. In fact, this no-go result

motivated the community to find a way to redefine the scenario so as to retrieve the quantum gain. It was commonly accepted that the most general scenario for quantum sensing should allow using arbitrary ancillary systems and arbitrary control operations over the entire system. It turned out that within such scenario the machinery of quantum error correction, coming from the field of quantum computing, could be successfully adapted, which partially resolved the problem (the most important work in this subject is the Ref. [30]). However, the answer to the general question under which conditions the classical scaling of precision in the most general metrological scenario can be surpassed was still missing. After this introductory remarks, I confirm that to the best of my knowledge the submitted manuscript solves completely this question in the case of Markovian noise. Completeness of the solution means that it provides a necessary and sufficient condition for the possibility of achieving the quantum gain given the Hamiltonian encoding the parameter and the description of noise (Lindblad operators). Moreover the proof of this fact is constructive, namely if the necessary and sufficient condition is met, the current work provides a construction of an optimal error correcting protocol, which allows for the HL scaling. I have no doubts that the work would be interesting for the entire quantum information and quantum foundations community, since it solves an open issue concerning the very existence of a quantum gain in one of the four main branches of quantum technologies (the other being communication, computation and simulation).

Therefore I might strongly support the publication of the submitted manuscript in Nature Communications on condition that the following issue is fully clarified. As already mentioned in an additional note, a similar work by R. Demkowicz-Dobrzanski et.al (cited as ref. [43]) was recently published (actually it appeared on arXiv almost 2 months before the current manuscript). Half of the work [43] concerns the same problem, namely possibility of achieving HL scaling of precision under general Markovian noise and with the help of quantum error correction. When it comes to details, the work [43] contains almost identical proof of the fact, that a violation of what is called in the current text the HNLS condition implies a linear scaling in time of the Quantum Fisher Information. Further in [43] the possibility of constructing a QEC (Quantum Error Correction) code in the case HNLS is fulfilled is discussed, however a general construction of such a code is left as an open question. Both of the discussed works, the current under revision, and [43] are generalizations of the work [30] that defines the scenario for the problem (using fast control operations to perform QEC in most general metrological protocol). In my opinion the current work deserves to be published in Nature Communications, assuming that it provides a standalone generalization of the work [30] not inspired by the partial results of [43]. Whereas I personally do not have any reason to doubt, that the current work, and the work [43] were being prepared completely independently, in my opinion such a statement, if true, should be enclosed in the current manuscript so as to avoid any sort of confusion among the community.

Although the current manuscript is generally well written, several issues should be concerned by the authors before publication.

The introduction section is concise and adequate, except for the second paragraph concerning the problem of precision limits in quantum metrology. The last sentence of the paragraph saying that for some types of noise the asymptotic precision is limited to the SQL insufficiently emphasises the issue. Series of works by R. Demkowicz-Dobrzanski (mainly Ref. [17]) demonstrated it to be almost impossible to achieve the asymptotic HL scaling for any reasonable realistic models of noise. This fact motivated redefinitions of metrological scenarios to avoid this conclusion. The work cited as [13] in the current manuscript, which introduced the idea of optimization of the probing time, was the first one to find a way to escape from the no-go result of [17] (in the current manuscript this work [13] is cited in the context of achieving HL scaling in noiseless systems, which seems not very adequate). Further it turned out that possibly the better option would be to adapt QEC protocols instead. In my opinion, all these issues should be clarified in order to introduce the reader to the landscape of main difficulties in the field of quantum metrology.

In the Results section my concerns include the construction of subsections: „QEC code for HL scaling when HNLS holds” and „Code optimization”. In my opinion, these subsections are the most important ones in the entire work, therefore they should be especially precise. In the section „QEC code for HL scaling when HNLS holds” the authors present the QEC conditions for achieving HL scaling, and construct a two-dimensional QEC code, based on spectral decomposition of the Hamiltonian

projected outside the „Lindblad span”. The section „Code optimization” in the first part

involves justification why a two-dimensional QEC code is sufficient to provide a HL scaling, whereas its second part concerns the issue of further optimization of the code so as to provide the best scaling factor. In my opinion the first part of the section „Code optimization”, starting from „When the evolution is noiseless (...)” and ending at formula (23) should appear at the beginning of section „QEC code for HL scaling when HNLS holds”, since:

- (i) understanding why two-dimensional QEC code suffices is crucial for the entire construction,
- (ii) actually it has nothing to do with optimization of the code itself.

When it comes to the technical details, in my opinion, the QEC conditions (15-17) should be compared with the ones from the parallel work [43]. Namely, in [43] the respective conditions (25)-(27) are very similar, except for the fact that the evolution-non-triviality condition (25) is stronger there, demanding that the evolution must create coherences between the code states. In the work under consideration, the respective condition (17) is weaker, demanding that the effective Hamiltonian in the code-states basis is not proportional to identity. Although I agree that such an assumption is sufficient, this issue should be commented.

The Conclusions are clearly written and I have no further remarks concerning the way of presentation.

My final remark concerns the following issue. It turns out that in the metrological scenario endowed with QEC the SQL and HL asymptotic scalings are dichotomic. Namely, either the SQL scaling cannot be surpassed, or it exists a QEC code which assures achieving the HL scaling. This fact is interesting on its own, since in metrological scenarios other than the QEC-assisted, intermediate

scaling bounds are possible (like in the time-optimization scenario of [13]). I think it would be advisable to mention this facts.

Finally, I found a possibly confusing typo. On page 5, left column, 12 row, it is written:

$\mathcal{H}_S \otimes \mathcal{H}_A$, whereas it should read: $\mathcal{H}_P \otimes \mathcal{H}_A$.

Reviewer #3 (Remarks to the Author):

This manuscript describes necessary and sufficient conditions to achieve the Heisenberg limit using error correction, and how to achieve the Heisenberg limit in cases where it is possible. Achieving the Heisenberg limit using error correction has been considered in previous work, but this work provides a method for general systems. This is quite an interesting result, because it was long assumed that the Heisenberg limit could never be reached in the presence of noise. This work shows generally how the Heisenberg limit can be achieved despite the presence of noise. An example is given of a nonlinear optical Hamiltonian with photon loss, which is shown to be correctable. That is an intriguing example, because it is not possible to correct for photon loss with a standard linear Hamiltonian.

The drawback to this work is that the results which are new don't seem to be relevant to real-world systems. The error correction needed for the optical system with a nonlinear Hamiltonian, while possible in principle, is completely unlike anything that can be achieved in reality. The most plausible case would be with a qubit, but that case was already considered thoroughly in reference [30]. What is new here is the general result for higher dimensions. Without a real system that these results could plausibly be used for, this seems like an interesting theoretical result, but not one of sufficient significance for Nature Communications.

A general drawback to the method for achieving the Heisenberg limit using error correction is that it assumes everything else is completely error free. It is assumed that the ancilla is error free, measurements are error free, and the corrections are error free. In contrast, the usual assumption for error correction is that all parts of the system are subject to error, and the goal is to reduce the logical error rate below the physical error rate. It is also assumed that the measurements and corrections are performed instantaneously, and at an arbitrarily high repetition rate. These assumptions seem very artificial, and it is likely that the real-world limitations would mean that error correction would not provide any improvement.

A minor issue is equation (9), which is cited as being from reference [30], but there does not appear to be any corresponding result in that work.

Overall I am reluctant to recommend this work for publication in Nature Communications because of the very implausible physical assumptions used. It might be suitable if they could give better justifications.

We thank the Reviewers for reading of our paper and giving constructive comments. In the following we provide detailed response to the comments.

Reviewer #1:

We are glad that Reviewer #1 found our work “to be an excellent paper which provides an elegant and novel solution to an outstanding problem in quantum metrology.” We implemented the changes suggested by Reviewer #1 as detailed below:

- *On p.2 below Fig. 1, the authors mention previous work showing that HL scaling can be achieved for qubits with perpendicular noise using QEC. They cite Ref. [16] (Escher et al), but to my knowledge this work did not address QEC. Perhaps the authors meant to cite Ref. [30] here?*

The right references [26-29] addressing QEC has been added here.

- *In the same paragraph, the σ 's are introduced but never defined to be the Pauli operators.*

We have added a sentence: “(Here $\sigma_{x,y,z}$ denote the Pauli matrices.)”.

- *Below eq. (2), the authors refer to the number of jump operators in the master equation as the noise 'rank'. However, it is unclear to me whether this is consistent with terminology in other works. For example, in Ref. [30], it is the rank of the matrix appearing in the master equation dissipator (in the Pauli basis). Could the authors clarify whether this terminology is standard?*

We clarified that rank is “the smallest number of Lindblad operators needed to describe the channel.”, which is the same definition as in Ref. [39] (formerly Ref.[30]).

- *In the last paragraph of p. 2 and top of p.3, the authors argue that when the Hamiltonian is nonlinear in the estimated parameter, one can first obtain sufficient precision on the parameter such that estimation is effectively local and the dependence can be treated as linear, without affecting the asymptotic scaling. To my knowledge, the quantum Cramer-Rao bound eq. (1) can only be ensured to be tight for local estimation, so one would anyway need to work in this regime.*

The flawed sentences have been modified as following: “If $H(\omega)$ is not a linear function of ω , the coding scheme we describe below can be repeated many times if necessary, using our latest estimate of ω after each round to adjust the scheme used in the next round. By including in the protocol an inverse Hamiltonian evolution step $\exp(iH(\hat{\omega})dt)$ applied to the probe, where $\hat{\omega}$ is the estimated value of ω , we can justify the linear approximation when $\hat{\omega}$ is sufficiently accurate.” Note that it is necessary to compensate the zeroth-order term in the expansion of $H(\omega)$; otherwise the QFI will be different.

- *In the first paragraph below Thm. 1, the authors consider a qubit example with a Lindblad operator given by a complex linear combination of Pauli operators, and conclude that the real and imaginary parts of the vector defining the noise must be parallel for it to be correctible. This was also derived in Ref. [30], so a citation here would be in order.*

The citation is added.

- *A typo: 5 lines below eq. (6) 'correcting' should be 'corrected'.*

The typo is corrected.

- *The λ and μ appearing in eqs (16) and (17) are never defined.*

After eq. (16) we add “for some complex numbers λ_k and μ_{kj} .”

- *Below eq. (19), some more explanation of why eqs. (18) and (19) imply that the conditions in eq. (15) are satisfied would be good.*

A detailed explanation of why eqs. (19) and (20) imply that the conditions in eq. (16) are satisfied has been added after eq. (20).

- *Below eq. (22), the authors say that the Cramer-Rao bound can be saturated. I think this is true only asymptotically for local estimation.*

The sentence is modified: “we can estimate ω with a precision which saturates the Cramer-Rao bound asymptotically for large number of measurements, realizing HL scaling.”

- *Below eq. (35), the authors find that the quantum Fisher information scales with the photon number to the fourth power. This seems to me to be due to the nonlinearity of the Hamiltonian, which induces photon-photon interactions (it is quadratic in the photon number). The scaling is consistent with that found for quantum metrology with interactions in Phys. Rev. Lett. 98, 090401 (2007). The authors might want to comment on this connection.*

We’ve mentioned this connection by a comment “The scaling $\delta\omega \sim 1/\bar{n}^2$ of the optimal precision arises from the nonlinear boson-boson interactions in the Hamiltonian Eq. (37) [47].”.

Reviewer #2:

We thank the Reviewer for reviewing the history of the field and supporting of publication of our work. We detail the changes we made addressing the Reviewer's comments:

- *Whereas I personally do not have any reason to doubt, that the current work, and the work [43] were being prepared completely independently, in my opinion such a statement, if true, should be enclosed in the current manuscript so as to avoid any sort of confusion among the community.*

We've added the statement in the additional note: "During the preparation of this manuscript, we became aware of related work by R. Demkowicz-Dobrzanski et al. [53], which provided a similar proof of the necessary condition in Theorem 1 and an equivalent description of the QEC conditions Eq. (16) and Eq. (18). We and the authors of Ref. [53] obtained this result independently. Both our paper and Ref. [53] generalize results obtained earlier in Ref. [39]."

- *The introduction section is concise and adequate, except for the second paragraph concerning the problem of precision limits in quantum metrology. The last sentence of the paragraph saying that for some types of noise the asymptotic precision is limited to the SQL insufficiently emphasizes the issue.*

We've added a sentence summarizing previous efforts to go beyond the SQL in the presence of decoherence: "The quest for measurement schemes surpassing the SQL has inspired a variety of clever strategies, such as optimizing the probing time [19], monitoring the environment [20, 21], and exploiting non-Markovian effects [22–25]." and mentioned "Quantum error correction (QEC) is a particularly powerful tool for enhancing the precision of quantum metrology [26–31]." in the next paragraph.

- *In my opinion the first part of the section „Code optimization”, starting from „When the evolution is noiseless (...)” and ending at formula (23) should appear at the beginning of section „QEC code for HL scaling when HNLS holds”, since: (i) understanding why two-dimensional QEC code suffices is crucial for the entire construction, (ii) actually it has nothing to do with optimization of the code itself.*

We would like to clarify the role of equations (24-26). First, equations (16,18) have already demonstrated that a two-dimensional QEC code, if constructed, is sufficient to achieve HL scaling, without the need of equations (24-26). The statement based on equations (24-26) is needed to narrow down the search of optimized QEC code, which helps to exclude higher dimensional codes for optimization. Therefore, we believe that it is more appropriate to keep the statements of equations (24-26) in the section of "Code Optimization".

- *When it comes to the technical details, in my opinion, the QEC conditions (15-17) should be compared with the ones from the parallel work [43]. Namely, in [43] the respective conditions (15)-(17) are very similar, except for the fact that the evolution-non-triviality condition (17) is stronger there, demanding that the evolution must create coherences*

between the code states.

We think that these two QEC conditions are actually equivalent, because if the evolution is non-trivial then we can perform a linear transformation of the code basis so that the Hamiltonian has a nonzero off-diagonal element. Therefore, we mention there is “an equivalent description of the QEC conditions Eq. (16) and Eq. (18)” provided by Ref. [53] in the additional note.

- *It turns out that in the metrological scenario endowed with QEC the SQL and HL asymptotic scalings are dichotomic. Namely, either the SQL scaling cannot be surpassed, or it exists a QEC code which assures achieving the HL scaling. This fact is interesting on its own, since in metrological scenarios other than the QEC-assisted, intermediate scaling bounds are possible (like in the time-optimization scenario of [13]). I think it would be advisable to mention this fact.*

We’ve added a comment on the dichotomy in the introduction (page 2, right column): “Notably, in the quantum metrology scheme considered here, either SQL scaling cannot be surpassed or HL scaling is achievable via quantum coding; in contrast, intermediate scaling is possible in some other metrology scenarios [19].”

- *Finally, I found a possibly confusing typo. On page 5, left column, 12 row, it is written: $H_S \otimes H_A$, whereas it should read: $H_P \otimes H_A$.*

The typo is fixed.

Reviewer #3:

We address the Reviewer's concern regarding the practical relevance as detailed below.

- *The drawback to this work is that the results which are new don't seem to be relevant to real-world systems. The error correction needed for the optical system with a nonlinear Hamiltonian, while possible in principle, is completely unlike anything that can be achieved in reality. The most plausible case would be with a qubit, but that case was already considered thoroughly in reference [30]. What is new here is the general result for higher dimensions. Without a real system that these results could plausibly be used for, this seems like an interesting theoretical result, but not one of sufficient significance for Nature Communications.*

We respectfully disagree with the reviewer. Besides two-level systems, there are many physical systems that use multiple levels for quantum sensing, such as nitrogen-vacancy centers with triplet ground states, Rydberg atoms with many energy levels, collective modes of spin ensembles, or even bosonic systems. We believe our results can be directly applied in these physical systems. To illustrate this point, we have added two examples at the end of the “Qubit probe” section.

- *A general drawback to the method for achieving the Heisenberg limit using error correction is that it assumes everything else is completely error free. It is assumed that the ancilla is error free, measurements are error free, and the corrections are error free. In contrast, the usual assumption for error correction is that all parts of the system are subject to error, and the goal is to reduce the logical error rate below the physical error rate. It is also assumed that the measurements and corrections are performed instantaneously, and at an arbitrarily high repetition rate. These assumptions seem very artificial, and it is likely that the real-world limitations would mean that error correction would not provide any improvement.*

We agree that it is challenging to have reliable ancilla with fast and high-fidelity operation and measurement. First, we mentioned at the end of Page 1 that “We endow the experimentalist with these powerful tools because we wish to address, as a matter of principle, how effectively QEC can overcome the deficiencies of the noisy probe system.”, which provides a QEC-applying guideline for experimentalists in future. However, we would like to point out Ref [35], which has experimentally demonstrated improvement in quantum metrology with quantum error correction. We mentioned on Page 2 that “Indeed, our assumptions are reasonably well satisfied by NV centers in diamond [35], where sensing of a magnetic field by an electron spin can be enhanced using a quantum code which takes advantage of the long coherence time of a nearby (ancilla) nuclear spin.” Finally, we emphasize that the corrections and measurements should be repeated “at the time scales faster than the noise rate” (Page 1, right column), which is basically the same requirement as in other common quantum error correction scenarios and is therefore a reasonable assumption.

- *A minor issue is equation (9), which is cited as being from reference [30], but there does not appear to be any corresponding result in that work.*

We think that equation (10) is equivalent to equation (31) in Ref. [39] by taking x to be the square root of $\|\alpha_{dt}\|$ to minimize the RHS of equation (31) in Ref. [39].

Reviewers' comments:

Reviewer #2 (Remarks to the Author):

I confirm that changes made by Authors to the manuscript meet my expectations in a way sufficient to recommend the paper for publication in Nature Communications without further hesitation. However, just for the sake of clarity of presentation, I strongly recommend the Authors to take into account my remark about the section "Code optimization". Namely, the problem is the following. In response to my comments the Authors say: "First, equations (16,18) have already demonstrated that a two-dimensional QEC code, if constructed, is sufficient to achieve HL scaling". In my opinion this is not so obvious when reading the section "QEC code for HL scaling when HNLS hold". This section proves, that when HNLS holds, it is indeed possible to construct a two-dimensional QEC code, which restores the unitarity of evolution of probe-ancilla state (eq. 17) in a way that there exists a state which is nontrivially affected by the new effective evolution (eq. 18). I can agree, that it implicitly suggests that it would be possible to achieve the Heisenberg scaling by probing such a state, but it is far from being obvious when first reading the paper. This fact is explicitly shown in the beginning of the next section in eqs. (24)-(26), and that's why I suggested that it would be better to involve this part in the previous section. If nevertheless the authors are certain, that the HL scaling comes obviously from eqs. (16)-(18) it should be explained at least in a single sentence. Without this it is a bit difficult for a reader to understand the logical construction of the proof of theorem 1.

Reviewer #3 (Remarks to the Author):

The authors have addressed some of my concerns, but I am still not convinced that the results are relevant to real-world systems. In my previous report I pointed out that the authors considered the examples of qubits and optical systems; for qubits the results have already been given in prior work, and for optical systems the error correction needed is unrealistic. To be more specific, for optics the error correction would need to distinguish between the code space as given in Eq. (40) and the error space, but such a measurement does not look like anything that could be achieved in optics.

The authors responded that there are "many physical systems that use multiple levels for quantum sensing", and added some text on page 4 mentioning theoretical cases where their HNLS criterion applies. This response does not address my concern, which is that the proposed error correction in their example is not realistic. This is the only example given that is not already covered by previous work, and the error correction looks like it could not be performed at all, which is particularly problematic considering the assumption that the error correction can be performed arbitrarily fast

and with perfect accuracy. There should be some discussion of what would be needed in order to be able to do the error correction, and the level of difficulty.

Alternatively, if there is some other multilevel system where the error correction can be performed more easily, that should be given as an example (with citations to experiments using that multilevel system). It is not adequate to just name multilevel systems that can be used for quantum sensing, without giving any details or citations.

Otherwise this is very good work, but I am reluctant to recommend publication unless they better justify the physical relevance.

We thank the Reviewers for reading of our paper and giving constructive comments. In the following we provide detailed response to the comments.

Reviewer #2

We are glad that Reviewer #2 found “changes made by Authors to the manuscript meet my expectations in a way sufficient to recommend the paper for publication in Nature Communications without further hesitation”. We implemented the change suggested by Reviewer #2 as detailed below:

- *I strongly recommend the Authors to take into account my remark about the section "Code optimization". Namely, the problem is the following. In response to my comments the Authors say: "First, equations (16,18) have already demonstrated that a two-dimensional QEC code, if constructed, is sufficient to achieve HL scaling". In my opinion this is not so obvious when reading the section "QEC code for HL scaling when HNLS hold". This section proves, that when HNLS holds, it is indeed possible to construct a two-dimensional QEC code, which restores the unitarity of evolution of probe-ancilla state (eq. 17) in a way that there exists a state which is nontrivially affected by the new effective evolution (eq. 18). I can agree, that it implicitly suggests that it would be possible to achieve the Heisenberg scaling by probing such a state, but it is far from being obvious when first reading the paper.*

We've moved the first paragraph in the section “Code optimization” containing the expressions of QFI and corresponding optimal initial states to the section “QEC code for HL scaling when HNLS hold” to explicitly show how HL scaling can be achieved under conditions (1-3).

Reviewer #3:

We address the Reviewer's concern regarding the practical relevance as detailed below.

- *In my previous report I pointed out that the authors considered the examples of qubits and optical systems; for qubits the results have already been given in prior work, and for optical systems the error correction needed is unrealistic. To be more specific, for optics the error correction would need to distinguish between the code space as given in Eq. (40) and the error space, but such a measurement does not look like anything that could be achieved in optics.*

In case of the “Kerr effect with photon loss” example, at the end of this section, we've added an experiment scheme that is possible to be implemented based on existing experimental techniques. First, we observe that the ancilla qubit can be discarded which helps simplify the QEC procedure. Second, we point out that the QEC can be carried out using quantum non-demolition parity measurements and control pulses, which has already been demonstrated using microwave photons in superconducting devices (see Ref. [38-40]).

- *The authors responded that there are "many physical systems that use multiple levels for quantum sensing", and added some text on page 4 mentioning theoretical cases where their HNLS criterion applies. This response does not address my concern, which is that the proposed error correction in their example is not realistic. This is the only example given that is not already covered by previous work, and the error correction looks like it could not be performed at all, which is particularly problematic considering the assumption that the error correction can be performed arbitrarily fast and with perfect accuracy. There should be some discussion of what would be needed in order to be able to do the error correction, and the level of difficulty.*

We've explained in detail (at the end of Page 1 and the beginning of Page 2) the experimental realizations of our scheme, as well as challenges and progress involved in these implementations. First, we pointed out that there are two types of systems that is suitable for our purpose – (i) hybrid systems where ancillas have long coherence times or (ii) systems with additional layer of error correction to suppress the errors from noisy ancillas (explained further in revised Methods). Second, in terms of rate and accuracy of QEC, the requirement that QEC should be performed fast and accurately enough such that the probability that one error occurs in time interval dt is small can be quite challenging. But we would like to point out that quantum error correction has recently been demonstrated in superconducting circuits with microwave photons (see Ref. [38-40]) where QEC has reached the breakeven point, and single- and two-qubit logical operations can be performed. Furthermore, if the sensing could be performed within a decoherence free subspace, then there is no need for additional error correction, which could make sensing with encoding more feasible in the near future.

- *Alternatively, if there is some other multilevel system where the error correction can be performed more easily, that should be given as an example (with citations to experiments*

using that multilevel system). It is not adequate to just name multilevel systems that can be used for quantum sensing, without giving any details or citations.

We've added an example – a multi-qubit sensor with qubits at distinct spatial positions, where the signal and noise are parallel for each individual qubit, but the signal and noise depend on position in different ways. It shows that there can be many physical platforms beyond two-level systems which could benefit from our techniques.

REVIEWERS' COMMENTS:

Reviewer #3 (Remarks to the Author):

The authors have added some detail about how to achieve their scheme experimentally. These are quite interesting results, and I think the manuscript should now be suitable to publish.